# GFNet: Geometric Flow Network for 3D Point Cloud Semantic Segmentation

**Haibo Qiu**                                                                *hqiu2518@sydney.edu.au*
*School of Computer Science, The University of Sydney, Australia*

**Baosheng Yu**                                                             *baosheng.yu@sydney.edu.au*
*School of Computer Science, The University of Sydney, Australia*

**Dacheng Tao**                                                            *dacheng.tao@sydney.edu.au*
*School of Computer Science, The University of Sydney, Australia*

**Reviewed on OpenReview:** *https://openreview.net/forum?id=LSAAlS7Yts*

## Abstract

Point cloud semantic segmentation from projected views, such as range-view (RV) and bird's-eye-view (BEV), has been intensively investigated. Different views capture different information of point clouds and thus are complementary to each other. However, recent projection-based methods for point cloud semantic segmentation usually utilize a vanilla late fusion strategy for the predictions of different views, failing to explore the complementary information from a geometric perspective during the representation learning. In this paper, we introduce a geometric flow network (GFNet) to explore the geometric correspondence between different views in an align-before-fuse manner. Specifically, we devise a novel geometric flow module (GFM) to bidirectionally align and propagate the complementary information across different views according to geometric relationships under the end-to-end learning scheme. We perform extensive experiments on two widely used benchmark datasets, SemanticKITTI and nuScenes, to demonstrate the effectiveness of our GFNet for project-based point cloud semantic segmentation. Concretely, GFNet not only significantly boosts the performance of each individual view but also achieves state-of-the-art results over all existing projection-based models. Code is available at `https://github.com/haibo-qiu/GFNet`.

## 1 Introduction

3D point cloud analysis has drawn increasing attention from both academic and industrial communities, since the wide deployments of lidar sensors have made it possible to obtain abundant 3D point cloud data (Behley et al., 2019; Caesar et al., 2020). Compared to 2D images (*e.g.*, RGB images), a point cloud can capture precise structures of objects, thus providing a geometry-accurate perspective representation, intrinsically in line with the 3D real world. Point cloud semantic segmentation, aiming to assign a semantic label to each point, is fundamental to scene understanding, which enables intelligent agents to precisely perceive not only the objects but also the dynamically changing environment. Therefore, point cloud semantic segmentation plays a crucial role, especially in safety-critical applications such as autonomous driving (Li et al., 2020; Aksoy et al., 2020) and robotics (Li et al., 2019; Yang et al., 2020).

Unlike structural pixels in an image, a point cloud is a set of points represented by $(x, y, z)$ coordinates without a specific order, and extremely sparse for in-the-wild scenes. Hence, it is non-trivial to utilize off-the-shelf deep learning technologies on images for point cloud analysis. Recent point cloud segmentation methods usually address the above-mentioned sparse distributed issue from the perspectives of either voxelization, single/multi-view projections, or novel point-based operations. However, voxel-based methods mainly suffer from heavy computations while point-based operations struggle to efficiently capture the neighbour information, especially when dealing with large-scale outdoor scenes (Behley et al., 2019; Caesar et al., 2020).

With the great success of fully convolutional networks for image-based semantic segmentation (Chen et al., 2017a;b; Long et al., 2015; Zhao et al., 2017), projection-based methods have recently received increasing attention. Figure 1 illustrates two widely used projected views, *i.e.*, range-view (RV) (Milioto et al., 2019) and bird's-eye-view (BEV) (Zhang et al., 2020b). Single view based methods can only learn view-specific representations (Alonso et al., 2020; Cortinhal et al., 2020; Xu et al., 2020), failing to handle those occluded points during projection. For example, the RV in Figure 2 shows a occluded tail phenomenon (*i.e.*, the distant occluded points are assigned with the labels of near displayed points) in the red rectangle areas. To address this problem, recent methods resort to multi-view models to incorporate complementary information over different views, which usually deal with RV/BEV in sequence (Chen et al., 2020; Gerdzhev et al., 2021) or perform a vanilla late fusion (Alnaggar et al., 2021; Liong et al., 2020). However, existing methods fail to probe the intrinsically geometric connections of RV/BEV during the representation learning.

As we can see from Figure 1, to find the geometric correspondence between two views (the dash line), we can utilize the original point cloud as a bridge, *e.g.*, the transformation RV to BEV can be obtained from two transformations (the solid line): 1) from RV to point cloud; and 2) from point cloud to BEV. Inspired by this, we introduce a novel geometric flow network (GFNet) to simultaneously learn view-specific representations and explore the geometric correspondences between RV and BEV in an end-to-end learnable manner. Specifically, we first propose to adopt two branches to process RV and BEV inputs, where each branch follows an encoder-decoder architecture using a ResNet (He et al., 2016) as the backbone. We then devise a geometric flow module (GFM), which is then applied at multiple levels of feature representations, to bidirectionally align and propagate geometric information across two projection views, aiming to learn more discriminative representations. Figure 2 illustrates an example of propagating the information from BEV to RV which benefits handling those occluded points by RV projection. In addition, inspired by Kochanov et al. (2020), we also use KPConv (Thomas et al., 2019) at the top of GFNet to replace a KNN post-processing, thus making it easy to train the overall multi-view point cloud semantic segmentation pipeline in an end-to-end paradigm. The main contributions of this paper are summarized as follows:

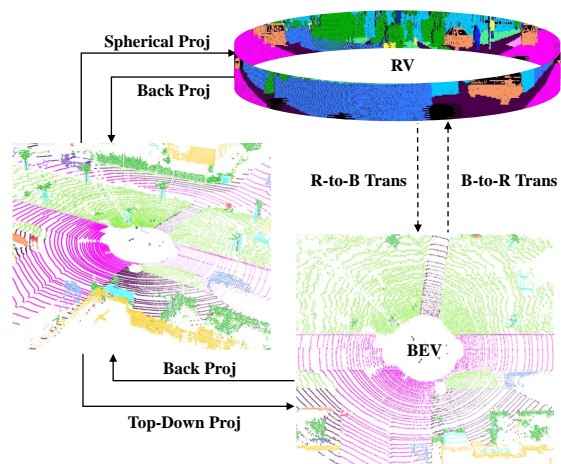

Figure 1: Geometric bidirectional transformation diagram between range-view (RV) and bird's-eye-view (BEV).

- We introduce a novel GFNet to simultaneously learn and fuse multi-view representations, where the proposed geometric flow module (GFM) enables the geometric correspondence information to flow across different views.

- We devise two branches for RV and BEV with KNN post-processing replaced by KPConv, making the proposed GFNet end-to-end trainable.

- Extensive experiments are performed on two popular large-scale point cloud semantic segmentation benchmarks, *i.e.*, SemanticKITTI and nuScenes, to demonstrate the effectiveness of GFNet, which achieves state-of-the-art performance over all existing projection-based models.

## 2 Related Work

In this section, we review recent point cloud semantic segmentation literature from the perspectives of point-based, voxel-based, and projection-based methods. In addition, we discuss more recently hybrid methods which simultaneously use multiple formats/modalities. Among all projection-based methods, we mainly focus on the multi-view projection-based methods.

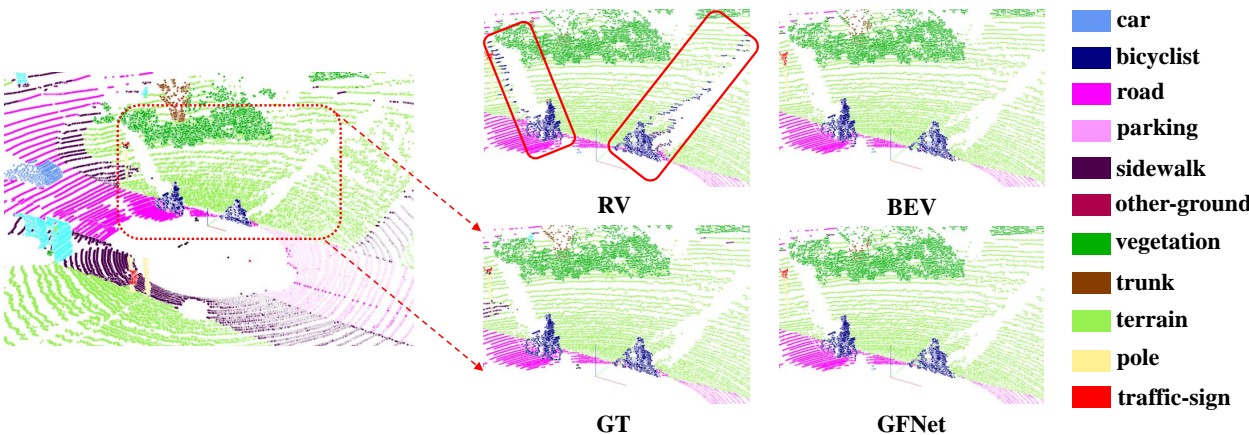

Figure 2: The distant occluded points caused by RV projection are misclassified as the labels of near displayed points in the red rectangle areas, while they are totally captured by BEV. By propagating the information between BEV and RV, this issue can be well addressed by our GFNet.

**Point-based Methods.** Recent methods mainly devise novel point operations/architectures to directly learn representations from raw points (Hu et al., 2020; Li et al., 2018; Qi et al., 2017a;b; Thomas et al., 2019; Wang et al., 2019; Tang et al., 2022), including mlp-based (Hu et al., 2020; Qi et al., 2017a;b), cnn-based (Li et al., 2018), and graph-based methods (Thomas et al., 2019; Wang et al., 2019). Specifically, PointNet (Qi et al., 2017a) is a pioneer that directly processes point cloud with multi-layer perceptron (MLP), which is improved by PointNet++ (Qi et al., 2017b) using a hierarchical neural network to learn local features. PointCNN (Li et al., 2018) learns a X-transformation from the input points for alignment, followed by typical convolution layers. DGCNN (Wang et al., 2019) proposes a new graph convolution module called EdgeConv to capture local geometric features. RandLA-Net (Hu et al., 2020) employs random point sampling with an effective local feature aggregation module to persevere the local information. KPConv (Thomas et al., 2019) introduces a new point convolution operator named Kernel Point Convolution to directly take neighbouring points as input and processes with spatially located weights. Nevertheless, the irregular and disordered characteristics of point clouds make it inefficient to capture the neighbour information.

**Voxel-based Methods.** They (Cheng et al., 2021; Tang et al., 2020; Yan et al., 2020a; Zhang et al., 2020a; Zhu et al., 2021) first voxelize point clouds to regular grids and process with 3D convolutions. Cylinder3D (Zhu et al., 2021) introduces the cylindrical partition and asymmetrical 3D convolution networks to tackle the issues of sparsity and varying density of point clouds. SPVNAS (Tang et al., 2020) proposes Sparse Point-Voxel Convolution (SPVConv), which is a lightweight 3D module consisting of the vanilla Sparse Convolution and the high-resolution point-based branch. Furthermore, 3D Neural Architecture Search (3D-NAS) is presented to obtain the efficient and effective architecture for semantic segmentation. AF2S3Net (Cheng et al., 2021) designs an AF2M to capture the global context and local details and an AFSM to learn inter-relationships between channels across multi-scale feature maps from AF2M. However, the distributions of large-scale outdoor scenes (*e.g.*, SemanticKITTI (Behley et al., 2019)) are extremely sparse, and the computations grow cubically when increasing the voxel resolution.

**Hybrid Methods.** Recent methods (Xu et al., 2021; Ye et al., 2021; Yan et al., 2022) usually focus on simultaneously using multiple formats/modalities (*e.g.*, voxel, points or natural images) to learn discriminative representations. DRINet++ Ye et al. (2021) proposes Sparse Feature Encoder to extract local context information from voxelized grids, and Sparse Geometry Feature Enhancement to enhance the geometric characteristics of sparse points using multi-scale sparse projection and attentive multi-scale fusion. RPVNet Xu et al. (2021) explores multiple and mutual information interactions among three views (*i.e.*, projection, voxel and point), following by a gated fusion module to adaptively merge the three features based on concurrent inputs. 2DPASS Yan et al. (2022) assists raw points with 2D natural images. It distills richer semantic and structural information from 2D images without strict paired data constraints to the pure 3D point network, by leveraging an auxiliary modal fusion and multi-scale fusion-to-single knowledge distillation (MSFSKD).

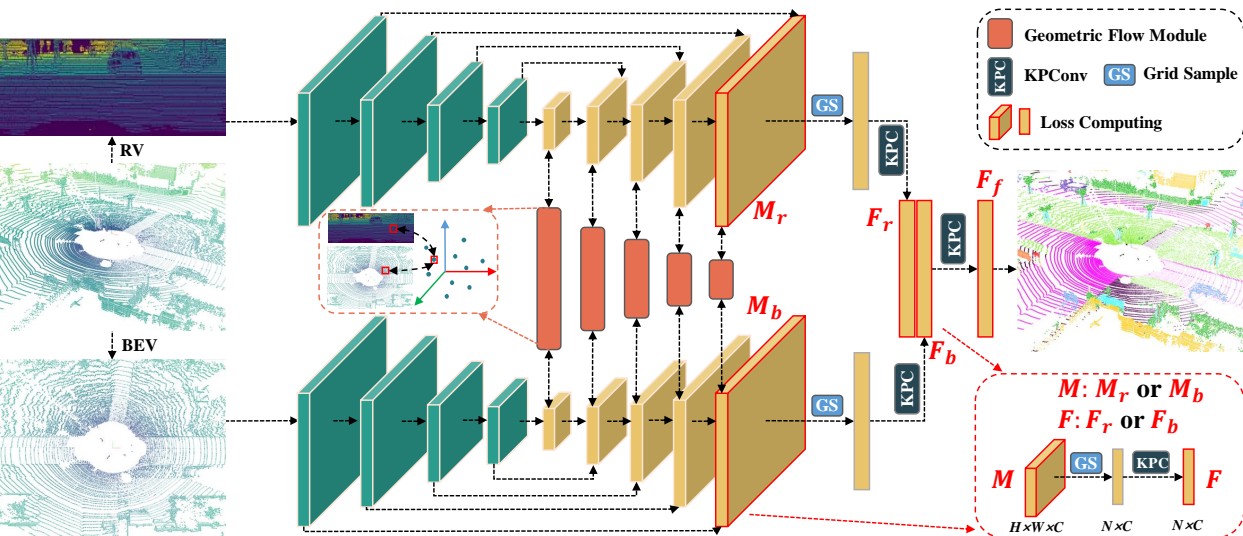

Figure 3: The overview of geometric flow network (GFNet). Point clouds are first projected to range-view (RV) and bird's-eye-view (BEV) using spherical and top-down projections, respectively. Then two branches with the proposed geometric flow module (GFM) handle RV/BEV to generate feature maps ($H \times W \times C$). Finally, grid sampling based on corresponding projection relationships is utilized to get the probability ($N \times C$) for each point, and the fused prediction $\mathbf{F_f}$ is obtained by applying kpconv on the concatenation of RV/BEV. Note that the subplot in bottom right corner illustrates how grid sampling works with dimension changing annotation, *i.e.*, sampling $\mathbf{F}$ ($\mathbf{F_r}$ or $\mathbf{F_b}$) with $N \times C$ from $\mathbf{M}$ ($\mathbf{M_r}$ or $\mathbf{M_b}$) with $H \times W \times C$.

**Projection-based Methods.** Point clouds are first projected to 2D images, *e.g.*, range-view (RV) (Alonso et al., 2020; Cortinhal et al., 2020; Milioto et al., 2019; Wu et al., 2018; 2019; Xu et al., 2020) and bird's-eye-view (BEV) (Zhang et al., 2020b), and then processed using 2D convolutions. For example, RangeNet++ (Milioto et al., 2019) adopts a DarkNet (Redmon & Farhadi, 2018) as the backbone to process RV images, and uses a KNN for post-processing. SqueezeSegV3 (Xu et al., 2020), standing on the shoulders of Wu et al. (2018; 2019), employs a spatially-adaptive Convolution (SAC) to adopt different filters for different locations according to input RV images. SalsaNext (Cortinhal et al., 2020) introduces a new context module which consists of a residual dilated convolution stack to fuse receptive fields at various scales. On the other hand, PolarNet (Zhang et al., 2020b) uses a polar-based birds-eye-view (BEV) instead of the standard 2D Cartesian BEV projections to better model the imbalanced spatial distribution of point clouds.

Among projection-based methods, applying multi-view projection can leverage rich complementary information (Alnaggar et al., 2021; Chen et al., 2020; Gerdzhev et al., 2021; Liong et al., 2020), while previous works usually process RV/BEV individually in sequence (Chen et al., 2020; Gerdzhev et al., 2021) or perform a vanilla late fusion (Alnaggar et al., 2021; Liong et al., 2020). For example, MVLidarNet (Chen et al., 2020) first obtains predictions from the RV image, which are then projected to BEV as initial features to learn representation by feature pyramid networks. Differently, TornadoNet (Gerdzhev et al., 2021) conducts in reverse order by devising a pillar-projection-learning module (PPL) to extract features from BEV, and then placing these features into RV, modeled by an encoder-decoder CNN. On the other side, MPF (Alnaggar et al., 2021) utilizes two different models to separately process RV and BEV, and then combines the predicted softmax probabilities from two branches as final predictions. AMVNet (Liong et al., 2020) takes a further step, *i.e.*, after obtaining the separate predictions from RV and BEV, it adopts a point head (Qi et al., 2017a) to refine the uncertain predictions, which are defined by the disagreements of two branches. Whereas, our GFNet enables geometric correspondence information to flow between RV/BEV at multi-levels during end-to-end learning, leading to a more discriminative representation and better performances.

# 3 Method

In this section, we first provide an overview of point cloud semantic segmentation and the proposed geometric flow network (GFNet). We then introduce projection-based point cloud segmentation using range-view (RV) and bird's-eye-view (BEV) in detail. After that, we describe the proposed geometric flow module (GFM), including geometric alignment and attention fusion. Lastly, the end-to-end optimization of GFNet is depicted.

## 3.1 Overview

Given a lidar point cloud with $N$ 3D points $\mathbf{P} \in \mathbb{R}^{N \times 4}$, we then have the format of each point as $(x, y, z, remission)$, where $(x, y, z)$ is the cartesian coordinate of the point relative to the lidar sensor and $remission$ indicates the intensity of returning laser beam. The goal of point cloud semantic segmentation is to assign all points in $\mathbf{P}$ with accurate semantic labels, *i.e.*, $\mathbf{Q} \in \mathbb{N}^N$. For projection-based point cloud semantic segmentation, we also need to transform the ground truth labels $\mathbf{Q}$ to the projected views during training, *i.e.*, $\mathbf{Q}_r$ for RV and $\mathbf{Q}_b$ for BEV.

The overall pipeline of GFNet is illustrated in Figure 3. Specifically, a point cloud $\mathbf{P}$ is first transformed to range-view (RV) as $\mathbf{I}_r$ and bird's-eye-view (BEV) as $\mathbf{I}_b$ using spherical and top-down projections, respectively. We then have two sub-network branches with encoder-decoder architectures to take RV/BEV images as inputs and generate dense predictions, which are referred to as the probability maps for each semantic class. The proposed geometric flow module (GFM) is incorporated into each layer of the decoder, bidirectionally propagating feature information according to the geometric correspondences across two views. After that, we obtain the classification probabilities of all points by applying a grid sampling on the dense probability maps, which is based on the projection relationship between a specific view and the original point cloud, as illustrated in the bottom right corner of Figure 3. Inspired by Kochanov et al. (2020), we also introduce KPConv (Thomas et al., 2019) on the top of the proposed GFNet to replace the KNN operation and capture the accurate neighbour information in a learnable way. By doing this, the overall multi-view point cloud semantic segmentation pipeline can be trained in an end-to-end manner.

## 3.2 Multi-View Projection

For projection-based methods, a point cloud $\mathbf{P} \in \mathbb{R}^{N \times 4}$ needs to be transformed to an image $\mathbf{I} \in \mathbb{R}^{HW \times C}$ first to leverage deep neural networks primarily developed for 2D visual recognition, where $H$ and $W$ indicate the spatial size of projected images and $C$ is the number of channels. Different projections are corresponding to different transformations, *i.e.*, $\mathcal{P} : \mathbb{R}^{N \times 4} \mapsto \mathbb{R}^{HW \times C}$. In this paper, we adopt two widely-used projected views for point cloud analysis, *i.e.*, range-view (RV) and bird's-eye-view (BEV). As shown in Figure 3, we aim to learn effective representations from two different views, RV and BEV, using the proposed two-branch networks with an encoder-decoder architecture in each branch. We describe the details of multi-view projection as follows.

**Range-View (RV).** To learn effective representations from RV images, spherical projection is required to first project a point cloud $\mathbf{P}$ to a 2D RV image (Milioto et al., 2019). Specifically, we first project a point $(x, y, z)$ from the cartesian space to the spherical space as follows:

$$\begin{bmatrix} \psi \\ \phi \\ r \end{bmatrix} = \begin{bmatrix} arctan(y, x) \\ arcsin(z/\sqrt{x^2 + y^2 + z^2}) \\ \sqrt{x^2 + y^2 + z^2} \end{bmatrix},$$
(1)

where $\psi, \phi$, and $r$ indicate azimuthal angle, polar angle, and radial distance (*i.e.*, the range of each point), respectively. We then have the pixel coordinate of $(x, y, z)$ in the projected 2D range image as

$$\begin{bmatrix} \widetilde{u} \\ \widetilde{v} \end{bmatrix} = \begin{bmatrix} (1 - \psi/\pi)/2 \cdot W \\ (f_{up} - \phi)/f \cdot H \end{bmatrix},$$
(2)

where $(H, W)$ represent the spatial size of range image, and $f = f_{up} - f_{down}$ is the vertical field-of-view of the lidar sensor. For each projected pixel $(u, v)$ (discretized from $(\widetilde{u}, \widetilde{v})$), we take the $(x, y, z, r, remission)$ as its feature, leading to a range image with the size of $(H, W, 5)$. In addition, an improved range-projected method is proposed by Triess et al. (2020), which further unfolds the point clouds following the captured order by the lidar sensor, leading to smoother projected images and a higher valid projection rate[1]. If not otherwise stated, we adopt this improved range projection (Triess et al., 2020) in all our experiments.

**Bird's-Eye-View (BEV).** To learn effective representations from BEV images, top-down orthogonal projection is employed to transform a point cloud into a BEV image (Chen et al., 2017c). Furthermore, the polar coordinate system is introduced to replace the cartesian system by Zhang et al. (2020b), which can be formulated as follows:

$$\begin{bmatrix} \widetilde{u} \\ \widetilde{v} \end{bmatrix} = \begin{bmatrix} \sqrt{x^2 + y^2} \cos(\arctan(y, x)) \\ \sqrt{x^2 + y^2} \sin(\arctan(y, x)) \end{bmatrix} = polar(x, y), \tag{3}$$

where $polar(\cdot)$ is the coordinate transformation from cartesian system to polar system. Following Zhang et al. (2020b), we use nine features to describe each pixel $(u, v)$ (by discretizing $(\widetilde{u}, \widetilde{v})$ to $[0, H - 1]$ and $[0, W - 1]$) in BEV image, including three relative cylindrical distance, three cylindrical coordinates, two cartesian coordinates and one remission, which can be constructed as follows:

$$[\Delta cylindrical(x, y, z), \ cylindrical(x, y, z), \ x, \ y, \ remission)], \tag{4}$$

where $cylindrical(x, y, z) = [polar(x, y), z]$ represents the cylindrical coordinates, $\Delta cylindrical(x, y, z)$ are the relative distances to the center of the BEV grid, and each BEV image thus has the shape of $(H, W, 9)$.

### 3.3 Geometric Flow Module

Intuitively, RV and BEV contain different view information of the original point cloud through different projections, leading to different information loss on different classes. For example, RV is good at those tiny or vertically-extended objects such as *motorcycle* and *person*, while BEV is sensitive to those objects with large and discriminative spatial size on the x-y plane. To sufficiently investigate the complementary information from RV/BEV, we explore them from a geometric perspective. Specifically, we devise a geometric flow module (GFM), which is based on the geometric correspondences between RV and BEV, to bidirectionally propagate the complementary information across different views. As illustrated in Figure 4, the first step is referred to as **Geometric Alignment**, which aligns the feature of source view (RV or BEV) to the target view using their geometric transformation; then the second step is called **Attention Fusion**, which applies the self-attention and the residual connection to combine the aligned feature representation with the original one. We describe the above-mentioned key steps of the proposed GFM module in detail as follows.

**Geometric Alignment.** The key idea lies in the geometric transformation matrices between two views, *i.e.*, $\mathbf{T}_{R \to B}$ (from RV to BEV) and $\mathbf{T}_{B \to R}$ (from BEV to RV). To obtain these transformation matrices, we propose to utilize the original point cloud as an intermediary agent. Specifically, from Eq.(1) and (2), we have the transformation from RV to the point cloud $\mathbf{P}$ as follows:

$$\mathbf{T}_{R \to P} = \begin{bmatrix} n_{0,0} & \cdots & n_{0,W_r-1} \\ \vdots & \vdots & \vdots \\ n_{H_r-1,0} & \cdots & n_{H_r-1,W_r-1} \end{bmatrix}, \tag{5}$$

where $\mathbf{T}_{R \to P} \in \mathbb{Z}^{H_r \times W_r}$, $(H_r, W_r)$ are the spatial size of 2D RV image, and $\{(n_{i,j}) | \ 0 <= i <= H_r - 1, \ 0 <= j <= W_r - 1\}$ is the $(n_{i,j})_{th}$ point which projects on $(i, j)$ coordinates. Note that if multiple points project to the same pixel, the point with smaller range is kept; If a pixel is not projected by any points, then its $n_{i,j}$ is assigned as $-1$. We then have the transformation from $\mathbf{P}$ to BEV image according to Eq.(3):

$$\mathbf{T}_{P \to B} = \begin{bmatrix} \mathbf{u}_0 & \cdots & \mathbf{u}_{N-1} \end{bmatrix}^T \tag{6}$$

$$= \begin{bmatrix} u_0 & \cdots & u_{N-1} \\ v_0 & \cdots & v_{N-1} \end{bmatrix}^T, \tag{7}$$

---

[1]Please refer to Appendix. A for more details.

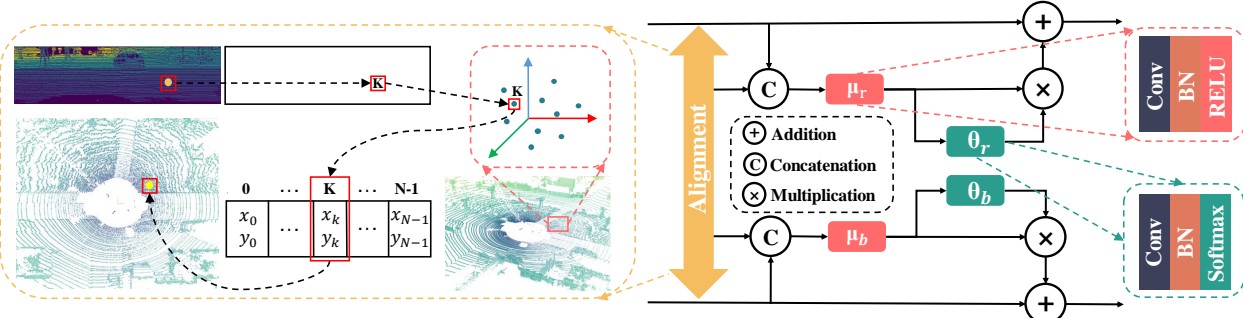

Figure 4: An overview of the proposed geometric flow module (GFM). It contains two main steps, *i.e.*, geometric alignment and attention fusion, which first aligns the feature from the source view (RV or BEV) to the target view using their geometric correspondences, and then applies self-attention and residual connections to combine view-specific features with the flowed information. Note that $\mu_r$ and $\theta_r$ share the same architecture but not weights with $\mu_b$ and $\theta_b$, respectively.

where $\mathbf{T}_{P \to B} \in \mathbb{Z}^{N \times 2}$, and $\{\mathbf{u}_k = (u_k, v_k) | \ 0 <= k <= N - 1\}$ are the projected pixel coordinates of 2D BEV image, corresponding to the $k_{th}$ point.

---

**Algorithm 1** Geometric Flow Module (BEV $\to$ RV)

---

**Input:** RV feature map $M_r : [H_r, W_r, C_r]$, BEV feature map $M_b : [H_b, W_b, C_b]$, $\mathbf{T}_{R \to B} : [H_r, W_r, 2]$.
**Output:** Fused RV feature map $M_{fused}^r : [H_r, W_r, C_r]$.

**Step 1: Geometric Alignment**

- Zero-initializing aligned feature $M_{b \to r}$ with the shape of $[H_r, W_r, C_b]$;
- **foreach** $(i, j) \in [1 : W_r] \times [1 : H_r]$ **do**
    $\mathbf{u} = \mathbf{T}_{R \to B}[i, j]$
    $u, v = \mathbf{u}$
    $M_{b \to r}[i, j, :] = M_b[u, v, :]$

**Step 2: Attention Fusion**

- Concatenating $M_r$ and $M_{b \to r}$ along the channel dimension as $M_{concat} : [H_r, W_r, C_r + C_b]$

- Applying the self-attention module to get $M_{atten} = \mu(M_{concat}) \cdot \theta(\mu(M_{concat}))$ with the shape of $[H_r, W_r, C_r]$

- Employing residual connection $M_{fused}^r = M_r + M_{atten}$

---

Lastly, we calculate the transformed matrix $\mathbf{T}_{R \to B}$ via $\mathbf{T}_{R \to P}$ and $\mathbf{T}_{P \to B}$. In particular, for each pixel $(i, j)$ in RV image, we first get its 3D point $n_{i,j} = \mathbf{T}_{R \to P}[i, j]$, then project $n_{i,j}$ to BEV image to obtain the corresponding pixel $\mathbf{T}_{P \to B}[n_{i,j}] = \mathbf{u}_{n_{i,j}}$. Now, we obtain $\mathbf{T}_{R \to B} \in \mathbb{Z}^{H_r \times W_r \times 2}$ as:

$$\mathbf{T}_{R \to B} = \begin{bmatrix} \mathbf{u}_{n_{0,0}} & \cdots & \mathbf{u}_{n_{0, W_r - 1}} \\ \vdots & \vdots & \vdots \\ \mathbf{u}_{n_{H_r - 1, 0}} & \cdots & \mathbf{u}_{n_{H_r - 1, W_r - 1}} \end{bmatrix}, \tag{8}$$

Once obtaining $\mathbf{T}_{R \to B}$, we can then align BEV features to RV features as follows: for each location $(i, j)$ in RV image, the $(u, v)$ coordinates in BEV image can be fetched via $\mathbf{T}_{R \to B}[i, j]$, and then we fuse the feature in $(u, v)$ to $(i, j)$ to get aligned feature $F_{b \to r}$. To align features from RV to BEV, we can operate in a similar way with $\mathbf{T}_{B \to R} \in \mathbb{Z}^{H_b \times W_b \times 2}$.

**Attention Fusion.** After the geometric feature alignment, we employ an attention fusion module to obtain the fused feature by concatenating the aligned feature and the target feature, which is followed by two convolution operations $\mu(\cdot)$ and $\theta(\cdot)$. They have simple architectures "Conv-BN-RELU" and "Conv-BN-Softmax" respectively, where the softmax function in $\theta$ is to map values to $[0, 1]$ as attention weights. The attention feature is finally combined with the target feature using a residual connection. We demonstrate the overall process of fusing BEV to RV, including geometric alignment and attention fusion modules, in Algorithm 1. The geometric flow from RV to BEV can be calculated similarly.

## 3.4 Optimization

Given $\mathbf{Q_r}$ as the labels for RV image $\mathbf{I_r}$ and $\mathbf{Q_b}$ for BEV image $\mathbf{I_b}$, which are projected from the original point cloud label $\mathbf{Q}$, we then have the 2D predictions $\mathbf{M_r}$ for RV and $\mathbf{M_b}$ for BEV, respectively. After that, we obtain the 3D predictions via grid sampling and KPConv, $i.e.$, $\mathbf{F_r}$ for RV and $\mathbf{F_b}$ for BEV. After fusion, we get the final 3D predictions $\mathbf{F_f}$. For simplicity and better illustration, we also highlight all predictions, $i.e.$, $\mathbf{M_r}, \mathbf{M_b}$ and $\mathbf{F_r}, \mathbf{F_b}, \mathbf{F_f}$, in Figure 3. To train the proposed GFNet, we first use the loss functions $\mathcal{L}_{2D}$ and $\mathcal{L}_{3D}$ for 2D and 3D predictions, respectively, as follows:

$$\mathcal{L}_{2D} = \rho \cdot \mathcal{L}_{CL}(\mathbf{M_r}, \mathbf{Q_r}) + \sigma \cdot \mathcal{L}_{CL}(\mathbf{M_b}, \mathbf{Q_b}), \tag{9}$$

and

$$\mathcal{L}_{3D} = \beta \cdot \mathcal{L}_{CE}(\mathbf{F_r}, \mathbf{Q}) + \gamma \cdot \mathcal{L}_{CE}(\mathbf{F_b}, \mathbf{Q}), \tag{10}$$

where $\mathcal{L}_{CE}$ indicates the typical cross entropy loss function while $\mathcal{L}_{CL}$ is the combination of the cross entropy loss and the Lovasz-Softmax loss (Berman et al., 2018) with weights $1:1$. We then apply the cross entropy loss on the final 3D predictions $\mathbf{F_f}$, that is, the overall loss function $\mathcal{L}_{total}$ can be evaluated as:

$$\mathcal{L}_{total} = \alpha \cdot \mathcal{L}_{CE}(\mathbf{F_f}, \mathbf{Q}) + \mathcal{L}_{3D} + \mathcal{L}_{2D}, \tag{11}$$

where $\lambda \doteq [\alpha, \beta, \gamma, \rho, \sigma]$ indicates the weight coefficient of different losses, and we investigate the influences of different loss terms in Sec. 4.4.

## 4 Experiments

In the section, we first introduce the adopted SemanticKITTI (Behley et al., 2019) and nuScenes (Caesar et al., 2020) datasets and the mean IoU and accuracy metric for point cloud segmentation. We then provide the implementation details of GFNet, including the network architectures and training settings. After that, we perform extensive experiments to demonstrate the effectiveness of GFM and analyze the influences of different hyper-parameters in GFNet. Lastly, we compare the proposed GFNet with recent state-of-the-art point/projection-based methods to show our superiority.

### 4.1 Datasets and Evaluation Metrics

**SemanticKITTI** (Behley et al., 2019), derived from the KITTI Vision Benchmark (Geiger et al., 2012), provides dense point-wise annotations for semantic segmentation task. The dataset presents 19 challenging classes and contains 43551 lidar scans from 22 sequences collected with a Velodyne HDL-64E lidar, where each scan contains approximately 130k points. Following Behley et al. (2019); Milioto et al. (2019), these 22 sequences are divided into 3 sets, $i.e.$, training set (00 to 10 except 08 with 19130 scans), validation set (08 with 4071 scans) and testing set (11 to 21 with 20351 scans). We perform extensive experiments on the validation set to analyze the proposed method, and also report performance on the test set by submitting the result to the official test server.

**nuScenes** (Caesar et al., 2020) is a large-scale autonomous driving dataset, containing 1000 driving scenes of 20 second length in Boston and Singapore. Specifically, all driving scenes are officially divided into training (850 scenes) and validation set (150 scenes). By merging similar classes and removing rare classes, point cloud semantic segmentation task uses 16 classes, including 10 foreground and 6 background classes. We use the official test server to report the final performance on test set.

**Evaluation Metrics.** Following Behley et al. (2019), we use mean intersection-over-union (mIoU) over all classes as the evaluation metric. Mathematically, the mIoU can be defined as:

$$mIoU = \frac{1}{C} \sum_{c=1}^{C} \frac{TP_c}{TP_c + FP_c + FN_c}, \tag{12}$$

where $TP_c$, $FP_c$, and $FN_c$ represent the numbers of true positive, false positive, and false negative predictions for the given class $c$, respectively, and $C$ is the number of classes. For a comprehensive comparison, we also report the accuracy among all samples, which can be formulated as:

$$Accuracy = \frac{TP + TN}{TP + FP + FN + TN}. \tag{13}$$

## 4.2 Implementation Details

For SemanticKITTI, we use two branches to learn representations from RV/BEV in an end-to-end trainable way, where each branch follows an encoder-decoder architecture with a ResNet-34 (He et al., 2016) as the backbone. The ASPP module (Chen et al., 2017b) is also used between the encoder and the decoder. The proposed geometric flow module (GFM) is incorporated into each upsampling layer. Note that the elements of $\mathbf{T}_{R \to B}, \mathbf{T}_{B \to R}$ fed into GFM are scaled linearly according to the current flowing feature resolution. For RV branch, point clouds are first projected to a range image with the resolution $[64, 2048]$, which is sequentially upsampled bilinearly to $[64 \times 2S, 2048 \times S]$ where $S$ is a scale factor. During training, a horizontal $1/4$ random crop of RV image, *i.e.*, $[128S, 512S]$, is used as data augmentation. On the other hand, we adopt polar partition (Zhang et al., 2020b) for BEV, and use a polar grid size of $[480, 360, 32]$ to cover a space of $[radius : (3m, 50m), z : (-3m, 1.5m)]$ relative to the lidar sensor. The grid first goes through a mini PointNet (Qi et al., 2017a) to obtain the maximum feature activations along the $z$ axis, leading to a reduced resolution $[480, 360]$ for BEV branch. We employ a SGD optimizer with momentum 0.9 and the weight decay $1e-4$. We use the cosine learning rate schedule (Loshchilov & Hutter, 2016) with warmup at the first epoch to 0.1. The backbone network is initialized using the pretrained weights from ImageNet (Deng et al., 2009). By default, we use $\lambda = [2.0, 2.0, 2.0, 1.0, 1.0]$ as the loss weight for Eq.11. We train the proposed GFNet for 150 epochs using the batch size 16 on four NVIDIA A100-SXM4-40GB GPUs with AMD EPYC 7742 64-Core Processor.

For nuScenes, we adopt Milioto et al. (2019) to project point clouds to a RV image with the resolution $[32, 1024]$ which is then upsampled bilinearly to $[32 \times 3S, 1024 \times S]$ where $S = 4$ in our experiments. Besides, a polar grid size of $[480, 360, 32]$ is used to cover a relative space of $[radius : (0m, 50m), z : (-5m, 3m)]$ for BEV branch. We train the model for total 400 epoch with batch size 56 using 8 NVIDIA A100-SXM4-40GB GPUs under AMD EPYC 7742 64-Core Processor. We adopt cosine learning rate schedule (Loshchilov & Hutter, 2016) with warmup at the first 10 epoch to 0.2. Other settings are kept the same with SemanticKITTI.

## 4.3 Effectiveness of GFM

In this part, we show the effectiveness of the proposed geometric flow module (GFM) as well as its influences on each single branch. As shown in Figure 3, we denote the results from $\mathbf{F_r}$ and $\mathbf{F_b}$ as *RV-Flow* and *BEV-Flow*, respectively, in regard to the information flow between RV and BEV brought by GFM. The predictions from $\mathbf{F_f}$ (obtained by applying KPConv on the concatenation of $\mathbf{F_r}$ and $\mathbf{F_b}$) are actually our final results, termed as *GFNet*. Note that the above results are evaluated using $\lambda = [2, 2, 2, 1, 1]$ for Eq.11. In addition, we train also each single branch separately without GFM modules, i.e., using $\lambda = [0, 2, 0, 1, 0]$ and $\lambda = [0, 0, 2, 0, 1]$ for *RV-Single* and *BEV-Single*, respectively.

We compare the performances of *RV/BEV-Single* and *BEV/BEV-Flow* in Table 1. Specifically, we find that both RV and BEV branches have been improved by a clear margin when incorporating with the proposed GFM module, e.g., 55.7% → 61.0% for BEV. Intuitively, RV is good at those vertically-extended objects like *motorcycle* and *person*, while BEV is sensitive to the classes with large and discriminative spatial size on the x-y plane. For example, *RV-Single* only achieves 32.4% on *truck* while *BEV-Single* obtains 64.8%, which is also illustrated by the first row of Figure 5 where RV predicts *truck* as a mixture of *truck, car*

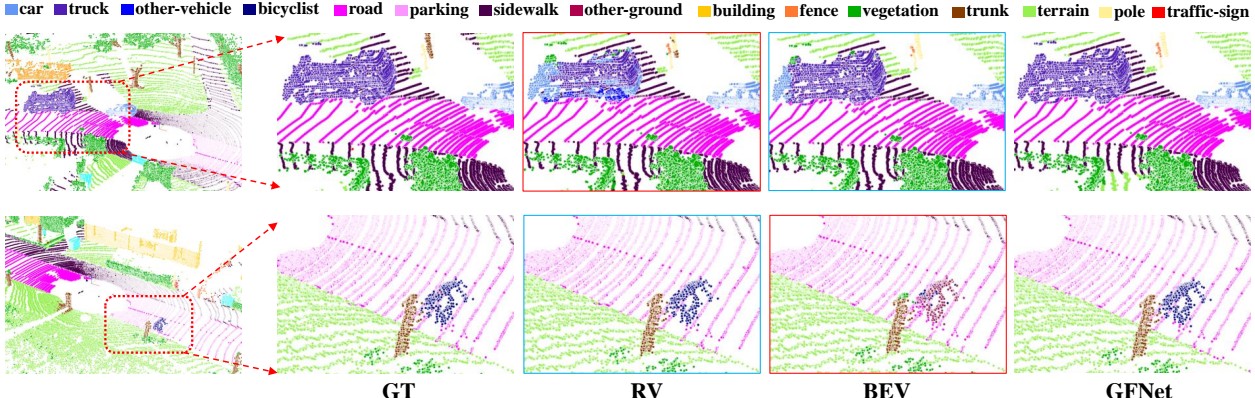

Figure 5: Visualization of RV and BEV. The view with the cyan contour helps the one with red. By incorporating both RV and BEV, our GFNet makes more accurate predictions.

and *other-vehicle*, but BEV acts much well. This is partially because *truck* is more discriminative on x-y plane (captured by BEV) than vertical direction (captured by RV) compared to *car, other-vehicle*. With the information flow from BEV to RV using GFM, *RV-Flow* significantly boosts the performance from 32.4% to 69.9%. A similar phenomenon can be observed in the second row of Figure 5, where BEV misclassifies *bicyclist* as *trunk*, since both of them are vertically-extended and also very close to each other, while RV predicts precisely. With the help of RV, *BEV-Flow* dramatically improves the performance from 55.7% to 61.0%. When further applying KPConv on the concatenation of *RV/BEV-Flow*, the proposed *GFNet* achieves the best performance 63.0%. These results demonstrate that the proposed GFM can effectively propagate complementary information between RV and BEV to boost the performance of each other, as well as the final performance.

Table 1: Quantitative comparisons in terms of mIoU to demonstrate the effectiveness of GFM on the validation set of SemanticKITTI.

| Method | car | bicycle | motorcycle | truck | other-vehicle | person | bicyclist | motorcyclist | road | parking | sidewalk | other-ground | building | fence | vegetation | trunk | terrain | pole | traffic-sign | mIoU |
|---|---|---|---|---|---|---|---|---|---|---|---|---|---|---|---|---|---|---|---|---|
| *RV-Single* | 93.7 | 48.7 | 57.7 | 32.4 | 40.5 | 69.2 | 79.9 | 0.0 | 95.9 | 53.4 | 83.9 | 0.1 | 89.2 | 59.0 | 87.8 | 66.1 | 75.3 | 64.0 | 45.2 | 60.1 |
| *RV-Flow* | 93.8 | 45.0 | 58.8 | 69.9 | 31.6 | 63.6 | 73.8 | 0.0 | 95.6 | 52.9 | 83.6 | 0.3 | 90.3 | 62.1 | 88.0 | 64.3 | 75.8 | 63.2 | 47.4 | **61.1** |
| *BEV-Single* | 93.6 | 29.9 | 42.4 | 64.8 | 26.8 | 48.1 | 74.0 | 0.0 | 94.0 | 45.9 | 80.7 | 1.4 | 89.2 | 46.5 | 86.9 | 61.4 | 74.9 | 56.8 | 41.6 | 55.7 |
| *BEV-Flow* | 93.7 | 43.7 | 61.2 | 74.0 | 31.0 | 61.6 | 80.6 | 0.0 | 95.3 | 53.1 | 82.8 | 0.2 | 90.8 | 61.4 | 88.0 | 63.1 | 75.6 | 58.9 | 43.1 | **61.0** |
| *GFNet* | 94.2 | 49.7 | 63.2 | 74.9 | 32.1 | 69.3 | 83.2 | 0.0 | 95.7 | 53.8 | 83.8 | 0.2 | 91.2 | 62.9 | 88.5 | 66.1 | 76.2 | 64.1 | 48.3 | **63.0** |

Table 2: Ablation studies of attention in GFM, loss weight coefficient $\lambda$ and scale factor $S$ on the SemanticKITTI val set.

(a) Attention in GFM

| attention | | mIoU |
|---|---|---|
| sigmoid | softmax | |
| | | 62.0 |
| ✓ | | 62.9 |
| | ✓ | 63.0 |

(b) $\lambda$ and $S$ under $\triangle = 1, \triangledown = 2$

| cfg | $\alpha$ | $\beta$ | $\gamma$ | $\rho$ | $\sigma$ | $S$ | mIoU |
|---|---|---|---|---|---|---|---|
| $a$ | $\triangle$ | | | | | 3 | 61.7 |
| $b$ | $\triangle$ | $\triangle$ | $\triangle$ | | | 3 | 61.8 |
| $c$ | $\triangle$ | $\triangle$ | $\triangle$ | $\triangle$ | $\triangle$ | 3 | 62.4 |
| $d$ | $\triangledown$ | $\triangledown$ | $\triangledown$ | $\triangle$ | $\triangle$ | 3 | 63.0 |
| $e$ | $\triangledown$ | $\triangledown$ | $\triangledown$ | $\triangle$ | $\triangle$ | 2 | 61.7 |
| $f$ | $\triangledown$ | $\triangledown$ | $\triangledown$ | $\triangle$ | $\triangle$ | 4 | 63.2 |

Table 3: Comparisons under mIoU, Accuracy and Frame Per Second (FPS) on SemanticKITTI test set. Note that the results of methods with * are obtained from RangeNet++ (Milioto et al., 2019). From top to down, the methods are grouped into point-based, projection-based and multi-view fusion models.

| Method | car | bicycle | motorcycle | truck | other-vehicle | person | bicyclist | motorcyclist | road | parking | sidewalk | other-ground | building | fence | vegetation | trunk | terrain | pole | traffic-sign | mIoU | Accuracy | FPS |
|---|---|---|---|---|---|---|---|---|---|---|---|---|---|---|---|---|---|---|---|---|---|---|
| PointNet* (Qi et al., 2017a) | 46.3 | 1.3 | 0.3 | 0.1 | 0.8 | 0.2 | 0.2 | 0.0 | 61.6 | 15.8 | 35.7 | 1.4 | 41.4 | 12.9 | 31.0 | 4.6 | 17.6 | 2.4 | 3.7 | 14.6 | - | 2 |
| PointNet++* (Qi et al., 2017b) | 53.7 | 1.9 | 0.2 | 0.9 | 0.2 | 0.9 | 1.0 | 0.0 | 72.0 | 18.7 | 41.8 | 5.6 | 62.3 | 16.9 | 46.5 | 13.8 | 30.0 | 6.0 | 8.9 | 20.1 | - | 0.1 |
| TangentConv* (Tatarchenko et al., 2018) | 86.8 | 1.3 | 12.7 | 11.6 | 10.2 | 17.1 | 20.2 | 0.5 | 82.9 | 15.2 | 61.7 | 9.0 | 82.8 | 44.2 | 75.5 | 42.5 | 55.5 | 30.2 | 22.2 | 35.9 | - | 0.3 |
| PointASNL (Yan et al., 2020b) | 87.9 | 0 | 25.1 | 39.0 | 29.2 | 34.2 | 57.6 | 0 | 87.4 | 24.3 | 74.3 | 1.8 | 83.1 | 43.9 | 84.1 | 52.2 | 70.6 | 57.8 | 36.9 | 46.8 | - | - |
| RandLa-Net (Hu et al., 2020) | 94.2 | 26.0 | 25.8 | 40.1 | 38.9 | 49.2 | 48.2 | 7.2 | 90.7 | 60.3 | 73.7 | 20.4 | 86.9 | 56.3 | 81.4 | 61.3 | 66.8 | 49.2 | 47.7 | 53.9 | 88.8 | 22 |
| KPConv (Thomas et al., 2019) | 96.0 | 30.2 | 42.5 | 33.4 | 44.3 | 61.5 | 61.6 | 11.8 | 88.8 | 61.3 | 72.7 | 31.6 | 90.5 | 64.2 | 84.8 | 69.2 | 69.1 | 56.4 | 47.4 | 58.8 | 90.3 | - |
| SqueezeSeg* (Wu et al., 2018) | 68.3 | 18.1 | 5.1 | 4.1 | 4.8 | 16.5 | 17.3 | 1.2 | 84.9 | 28.4 | 54.7 | 4.6 | 61.5 | 29.2 | 59.6 | 25.5 | 54.7 | 11.2 | 36.3 | 30.8 | - | 55 |
| SqueezeSegV2* (Wu et al., 2019) | 81.8 | 18.5 | 17.9 | 13.4 | 14.0 | 20.1 | 25.1 | 3.9 | 88.6 | 45.8 | 67.6 | 17.7 | 73.7 | 41.1 | 71.8 | 35.8 | 60.2 | 20.2 | 36.3 | 39.7 | - | 50 |
| RangeNet++ (Milioto et al., 2019) | 91.4 | 25.7 | 34.4 | 25.7 | 23.0 | 38.3 | 38.8 | 4.8 | 91.8 | 65.0 | 75.2 | 27.8 | 87.4 | 58.6 | 80.5 | 55.1 | 64.6 | 47.9 | 55.9 | 52.2 | 89.0 | 12 |
| PolarNet (Zhang et al., 2020b) | 93.8 | 40.3 | 30.1 | 22.9 | 28.5 | 43.2 | 40.2 | 5.6 | 90.8 | 61.7 | 74.4 | 21.7 | 90.0 | 61.3 | 84.0 | 65.5 | 67.8 | 51.8 | 57.5 | 54.3 | 90.0 | 16 |
| 3D-MiniNet-KNN (Alonso et al., 2020) | 90.5 | 42.3 | 42.1 | 28.5 | 29.4 | 47.8 | 44.1 | 14.5 | 91.6 | 64.2 | 74.5 | 25.4 | 89.4 | 60.8 | 82.8 | 60.8 | 66.7 | 48.0 | 56.6 | 55.8 | 89.7 | 28 |
| SqueezeSegV3 (Xu et al., 2020) | 92.5 | 38.7 | 36.5 | 29.6 | 33.0 | 45.6 | 46.2 | 20.1 | 91.7 | 63.4 | 74.8 | 26.4 | 89.0 | 59.4 | 82.0 | 58.7 | 65.4 | 49.6 | 58.9 | 55.9 | 89.5 | 6 |
| SalsaNext (Cortinhal et al., 2020) | 91.9 | 48.3 | 38.6 | 38.9 | 31.9 | 60.2 | 59.0 | 19.4 | 91.7 | 63.7 | 75.8 | 29.1 | 90.2 | 64.2 | 81.8 | 63.6 | 66.5 | 54.3 | 62.1 | 59.5 | 90.0 | 24 |
| MVLidarNet (Chen et al., 2020) | 87.1 | 34.9 | 32.9 | 23.7 | 24.9 | 44.5 | 44.3 | 23.1 | 90.3 | 56.7 | 73.0 | 19.1 | 85.6 | 53.0 | 80.9 | 59.4 | 63.9 | 49.9 | 51.1 | 52.5 | 88.0 | 92 |
| MPF (Alnaggar et al., 2021) | 93.4 | 30.2 | 38.3 | 26.1 | 28.5 | 48.1 | 46.1 | 18.1 | 90.6 | 62.3 | 74.5 | 30.6 | 88.5 | 59.7 | 83.5 | 59.7 | 69.2 | 49.7 | 58.1 | 55.5 | - | 21 |
| TORNADONet (Gerdzhev et al., 2021) | 94.2 | 55.7 | 48.1 | 40.0 | 38.2 | 63.6 | 60.1 | 34.9 | 89.7 | 66.3 | 74.5 | 28.7 | 91.3 | 65.6 | 85.6 | 67.0 | 71.5 | 58.0 | 65.9 | 63.1 | 90.7 | 4 |
| AMVNet (Liong et al., 2020) | 96.2 | 59.9 | 54.2 | 48.8 | 45.7 | 71.0 | 65.7 | 11.0 | 90.1 | 71.0 | 75.8 | 32.4 | 92.4 | 69.1 | 85.6 | 71.7 | 69.6 | 62.7 | 67.2 | 65.3 | 91.3 | - |
| GFNet (ours) | 96.0 | 53.2 | 48.3 | 31.7 | 47.3 | 62.8 | 57.3 | 44.7 | 93.6 | 72.5 | 80.8 | 31.2 | 94.0 | 73.9 | 85.2 | 71.1 | 69.3 | 61.8 | 68.0 | 65.4 | 92.4 | 10 |

## 4.4 Ablation Studies

In this subsection, we first explore the impacts of attention mechanism in GFM, the loss weights $\lambda$ defined in Eq.11; and the scale factor $S$ introduced in Sec. 4.2. In the default setting, we use the softmax attention with $\lambda = [2, 2, 2, 1, 1]$ and $S = 3$.

As shown in Table 2a, without attention mechanism (*i.e.*, no $\theta(\cdot)$ and $\otimes$ in Figure 4), the performance 62.0% is obviously inferior to the counterparts 62.9% and 63.0%, indicating that the attention operation helps to focus on the strengths instead of weaknesses of source view when fusing it into target view. If not otherwise stated, we use the softmax attention in our experiments. We evaluate the influences of $\lambda \doteq [\alpha, \beta, \gamma, \rho, \sigma]$ in Table 2b, where $\triangle = 1, \triangledown = 2$, e.g., we have $\lambda \doteq [\alpha, \beta, \gamma, \rho, \sigma] = [2.0, 2.0, 2.0, 1.0, 1.0]$ the configuration $d$. Specifically, when comparing the configurations $a$ to $b$ and $c$, we see that that additional supervisions on dense 2D and each branch RV/BEV 3D predictions further improve model performance. When comparing $c$ and $d$, a large weight on 3D prediction brings a better result. Therefore, if not otherwise stated, we adopt $\lambda = [2.0, 2.0, 2.0, 1.0, 1.0]$ for remaining experiments. The scale factor $S$ in Sec. 4.2 indicates the resolution of RV image, e.g., when $S = 3$, we have $[128S, 512S] = [384, 1536]$ and $[128S, 2048S] = [383, 6144]$ for training and testing, respectively. When comparing $d$ and $e$ in Table 2b, we find that a higher resolution significantly improves model performance, from 61.7% to 63.0%, further enlarging $S$ from 3 to 4 only brings a slightly better performance. For a better speed-accuracy tradeoff, we use $S = 3$ in the default setting.

## 4.5 Comparison with Recent State-of-the-Arts

**SemanticKITTI.** For fair comparison with recent methods, we follow the same setting in Behley et al. (2019); Kochanov et al. (2020), *i.e.*, both training and validation splits are used for training when evaluating on the test server. As shown in Table 3, GFNet achieves the new state-of-the-art performance 65.4% mIoU, significantly surpassing point-based methods (*e.g.*, 58.8% for KPConv (Thomas et al., 2019)) and single view models (*e.g.*, 59.5% for SalsaNext (Cortinhal et al., 2020)). For multi-view approaches (Alnaggar et al., 2021; Chen et al., 2020; Gerdzhev et al., 2021; Liong et al., 2020), GFNet outperforms recent methods Alnaggar et al. (2021); Chen et al. (2020); Gerdzhev et al. (2021) by a large margin. Comparing with AMVNet Liong et al. (2020), GFNet clearly outperforms it on the point-wise accuracy, i.e., 92.4% *vs.* 91.3%. The superior performance of GFNet shows the effectiveness of bidirectionally aligning and propagating geometric information between RV/BEV. Notably, AMVNet requires to train models for RV/BEV branch as well as their post-processing point head separately, while GFNet is end-to-end trainable. Additionally, since the acquisition frequency of the Velodyne HDL-64E LiDAR sensor (used by SemanticKITTI) is 10 Hz, GFNet can thus run in real-time, i.e., 10 FPS.

Table 4: Comparisons on nuScenes (the test set) under mIoU and Frequency Weighted IoU (or FW IoU).

| Method | barrier | bicycle | bus | car | const-vehicle | motorcycle | pedestrian | traffic-cone | trailer | truck | dri-surface | other-flat | sidewalk | terrain | manmade | vegetation | mIoU | FW IoU |
|---|---|---|---|---|---|---|---|---|---|---|---|---|---|---|---|---|---|---|
| PolarNet (Zhang et al., 2020b) | 72.2 | 16.8 | 77.0 | 86.5 | 51.1 | 69.7 | 64.8 | 54.1 | 69.7 | 63.4 | 96.6 | 67.1 | 77.7 | 72.1 | 87.1 | 84.4 | 69.4 | 87.4 |
| AMVNet (Liong et al., 2020) | 79.8 | 32.4 | 82.2 | 86.4 | 62.5 | 81.9 | 75.3 | 72.3 | 83.5 | 65.1 | 97.4 | 67.0 | 78.8 | 74.6 | 90.8 | 87.9 | 76.1 | 89.5 |
| GFNet (ours) | 81.1 | 31.6 | 76.0 | 90.5 | 60.2 | 80.7 | 75.3 | 71.8 | 82.5 | 65.1 | 97.8 | 67.0 | 80.4 | 76.2 | 91.8 | 88.9 | 76.1 | 90.4 |

**nuScenes.** To evaluate the generalizability of GFNet, we also report the performance on the testset in Table 4 by submitting results to the test server. Similarly, GFNet achieves superior mIoU performance 76.1%, which remarkably outperforms PolarNet (Zhang et al., 2020b) and tights AMVNet (Liong et al., 2020). However, our result 90.4% under Frequency Weighted IoU (FW IoU) beats 89.5% from AMVNet (Liong et al., 2020), which is consistent with the accuracy comparison on SemanticKITTI. It also reveals that GFNet performs much better on frequent classes while somewhat struggles on those rare/small classes. Despite the good performance of GFNet, it is also interesting to further improve GFNet by addressing rare/small classes from the perspectives of data sampling/augmentation and loss function.

## 5 Conclusion

In this paper, we propose a novel geometric flow network (GFNet) to learn effective view representations from RV and BEV. To enable propagating the complementary information across different views, we devise a geometric flow module (GFM) to bidirectionally align and fuse different view representations via geometric correspondences. Additionally, by incorporating grid sampling and KPConv to avoid time-consuming and non-differentiable post-processing, GFNet can be trained in an end-to-end paradigm. Extensive experiments on SemanticKITTI and nuScenes confirm the effectiveness of GFM and demonstrate the new state-of-the-art performance on projection-based point cloud semantic segmentation.

There are some limitations of the proposed method, since it builds upon two specific point cloud projection methods. Specifically, both RV and BEV may not be applicable to indoor datasets such as S3DIS (Armeni et al., 2016). For example, in a indoor scene of the bookcase, common objects such as table and chair are distinguishable and meaningful in the vertical direction, while the height information is missing for BEV. Also, RV image requires a scan cycle by the lidar sensor, which typically appears in outdoor scenarios such as autonomous driving (Please also refer to Appendix. B for more details and figures). Additionally, we apply the proposed geometric flow module in each decoder layer (*i.e.*, the upsampling layers), while popular point cloud object detection frameworks don't have such a decoder structure. Therefore, it is also non-trivial to directly apply the proposed method for object detection, which will be the subject of future studies.

## 6 Acknowledgement

This work is partially supported by ARC project FL-170100117.

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

# A    RV Projection

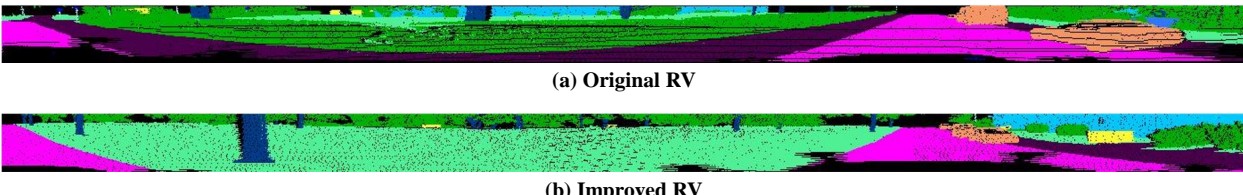

**(a) Original RV**

**(b) Improved RV**

Figure 6: Range images from original RV (Milioto et al., 2019) and improved RV (Triess et al., 2020). As we can see, Triess et al. (2020) obtains smoother projected image than Milioto et al. (2019).

We project 3D point cloud **P** to a 2D RV image with the size of $(H, W)$: due to the 2D-to-3D ambiguity, there are pixels that are not projected by any points, while multiple points might be projected to the same pixel. We define the valid projection rate as the ratio of valid pixels (*i.e.*, projected by as least one point) comparing to total pixels $HW$. Specifically, more valid pixels usually result in a smoother projected

Table 5: Valid projection rate (%) when using two different RV projections (Milioto et al., 2019; Triess et al., 2020) to generate images of $64 \times 2048$ size.

| Method | Train | Val |
|---|---|---|
| Original RV (Milioto et al., 2019) | 72.47 | 72.12 |
| Improved RV (Triess et al., 2020) | **83.69** | **83.51** |

image, *i.e.*, the higher valid projection rate, the better. We compare two different projection methods (Milioto et al., 2019; Triess et al., 2020) in terms of valid projection rate in 5. As we can observe, Triess et al. (2020) significantly outperforms Milioto et al. (2019) by over 11% in both train and val set. In addition, we visualize the RV images obtained by Milioto et al. (2019); Triess et al. (2020) separately in Figure 6. Obviously, the RV image generated by Triess et al. (2020) is clearly smoother than Milioto et al. (2019). Therefore, we use Triess et al. (2020) in all experiments if not states otherwise.

# B    RV under Indoor and Outdoor Scenes

In this section, we first described the RV projection (*i.e.*, spherical projection) in detail. We then show the difference between the point clouds collected in outdoor and indoor scenes. Lastly, we discuss why RV projection is not suitable for indoor scenes.

As shown in Figure 7, given a 3D point $p$, we first obtain the corresponding $\psi, \phi$ for spherical projection. We then normalize $\phi, \psi$ to $[0, 1]$ and map them to 2D RV image with size $[H, W]$ according to Eq. 2. However, point clouds in outdoor and indoor scenes are usually collected in different ways. For example, SemanticKITTI is collected via a Velodyne HDL-64E lidar on the top of the driving car, which launches lasers to all-around $(360°)$ horizontal directions and a certain degree $[f_{down}, f_{up}]$ vertical directions. When applied RV projection, a meaningful projected cylindrical image can be obtained (please refer to Figure 1). But for indoor dataset like S3DIS (Armeni et al., 2016), it scans the entire room in any directions with a Matterport (Inc., 2015) scanner to generate point clouds. In addition, those indoor objects are much more dense than outdoor objects. If we still want to use RV projection, it will lead to a

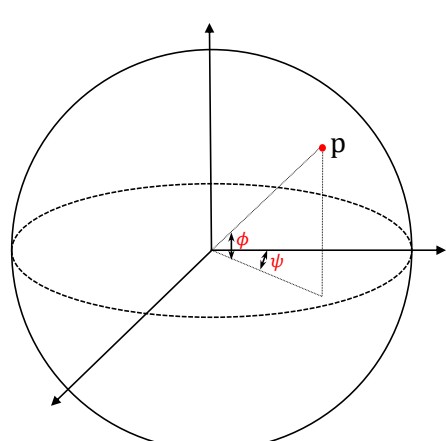

Figure 7: Range-view projection (*i.e.*, spherical projection). $p$ is a 3D point, $\psi, \phi$ are the azimuthal angle, polar angle of $p$.

severe distort image. We have also made some attempts using RV projection for S3DIS, but obtained meaningless images as in Figure 9. That is also the reason why existing projection-based methods (Milioto et al., 2019; Wu et al., 2018; Cortinhal et al., 2020) only employ RV projection in outdoor lidar-collected point clouds. As for indoor datasets like S3DIS, the mainstream methods (Qi et al., 2017a;b; Thomas et al., 2019) usually take raw points as input directly given that their size is much smaller than outdoor scenes.

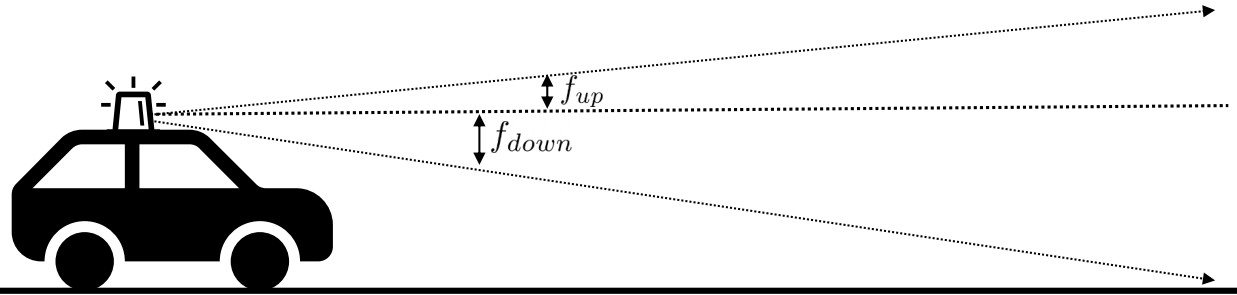

Figure 8: A simple diagram of the working mechanism when the lidar sensor collects point clouds. The lidar launches lasers to all-around ($360°$) horizontal directions and a certain degree $[f_{down}, f_{up}]$ vertical directions. Note that the vertical field of view $f = f_{up} - f_{down}$ where $f_{down}$ is negative.

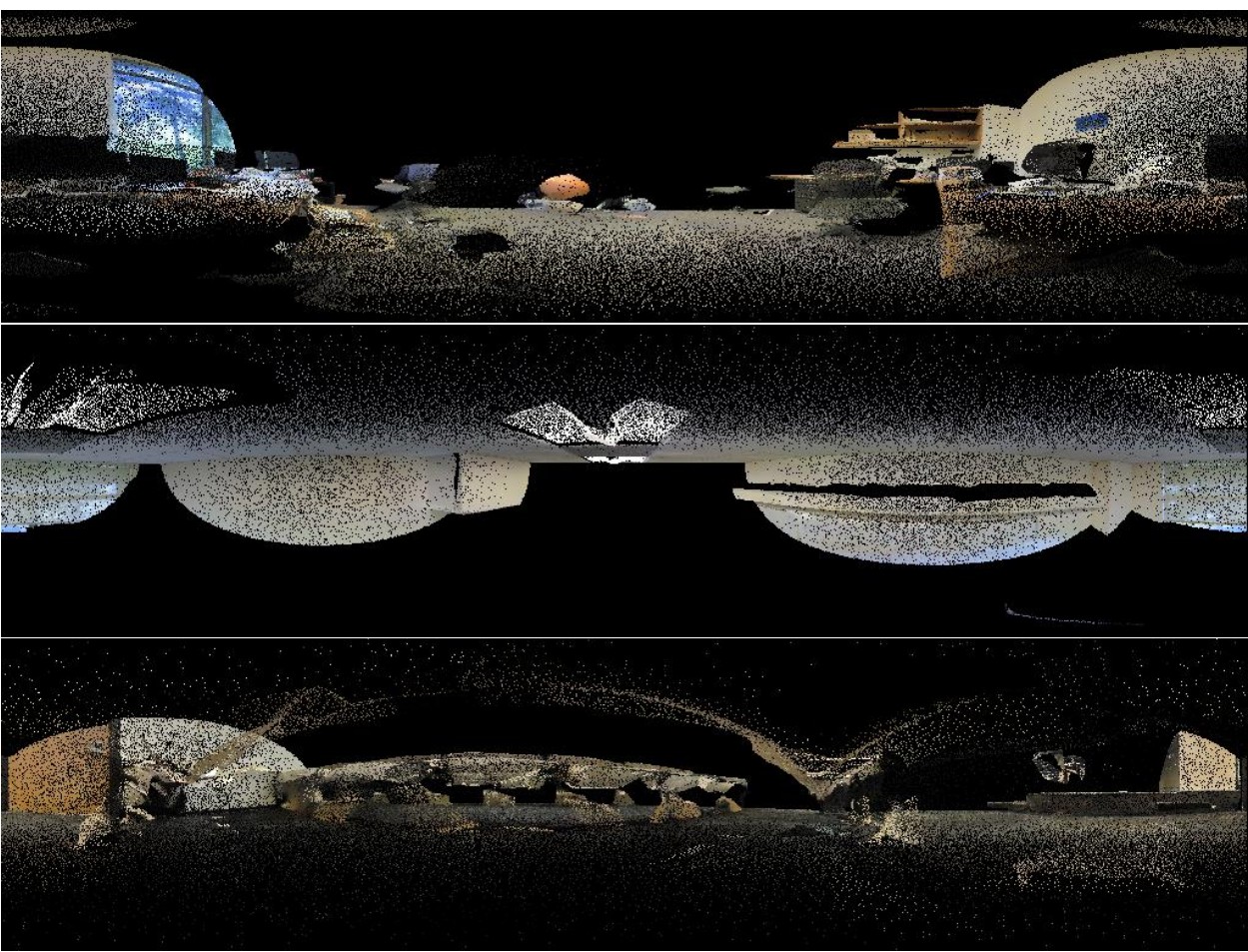

Figure 9: Three visualizations of samples from S3DIS (Armeni et al., 2016) using RV projection.

## C   Visualization

Figure 10 illustrates comparisons between GFNet and ground truth on complex scenes, revealing the excellent performance of GFNet. We also provide a GIF image, *i.e.*, `figs/vis.gif`, at `https://github.com/haibo-qiu/GFNet` for more visualizations.

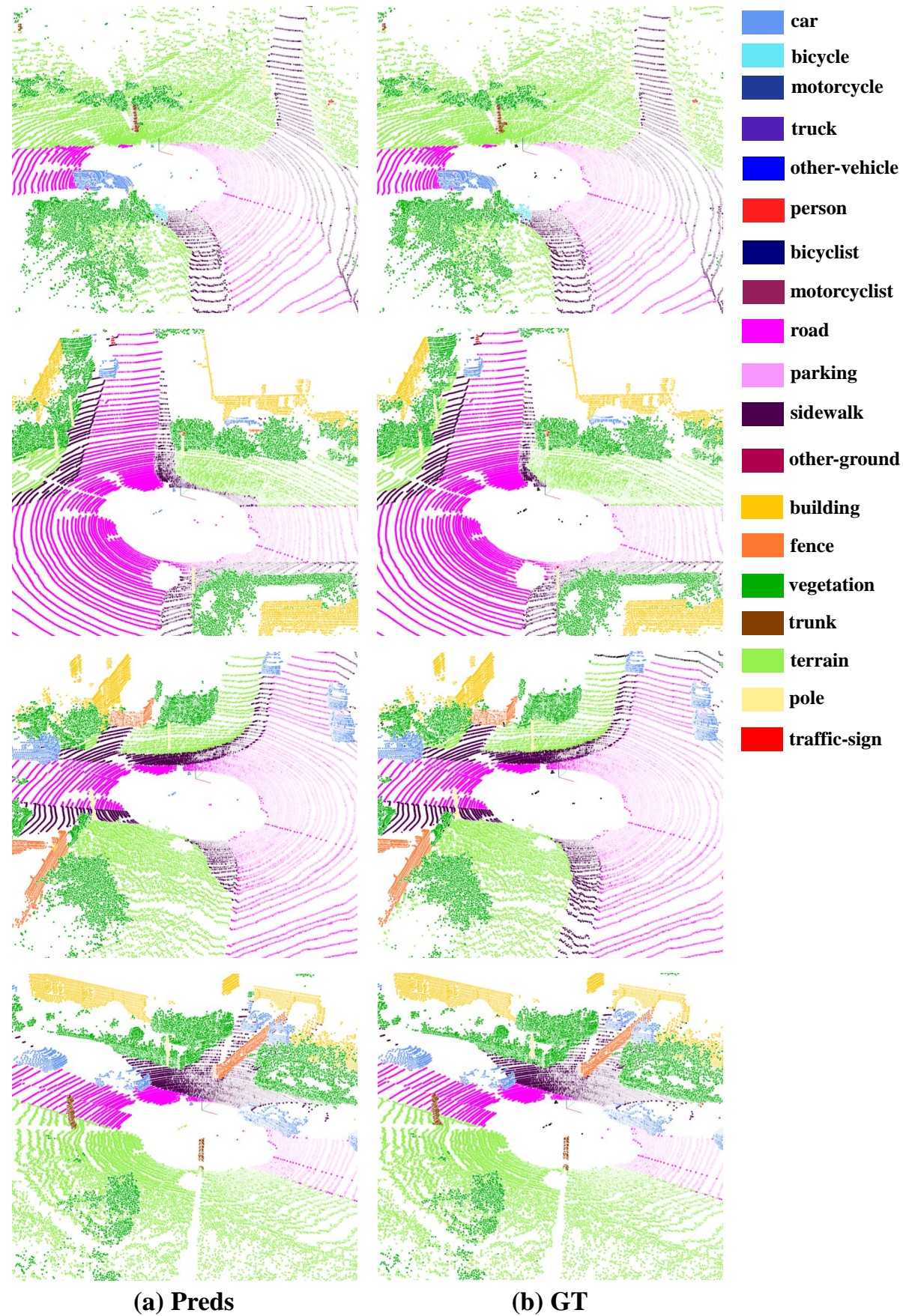

**(a) Preds**       **(b) GT**

Figure 10: Visualization of the predictions from our GFNet comparing to GT.

