# OpenReview forum: "GFNet: Geometric Flow Network for 3D Point Cloud Semantic Segmentation"
_TMLR — Accepted by TMLR_

### Review · Reviewer_m9mP · 2022-07-21

**Summary Of Contributions:**

A method for 3D pointcloud semantic segmentation is proposed. The core idea is to operate at two projected representations of the input 3D pcl: BEV and RV. At each decoder layer the two branches share information through a simple geometric alignment module that pushes features from each branch to the corresponding place on the other branch. Finally, both branches are combined and jointly processed in 3D to get the final prediction. Experiments show good overall performance with consistent improvement over using only BEV or RV or 3D.

**Requested Changes:**

The conceptual and technical contribution of the work is rather small. Yet, performance is impressive and therefore opens up interesting questions. Aside for addressing all the clarifications asked under "Strengths And Weaknesses" i've also listed several ideas for improving the manuscript like adding tasks, testing different projections, discussing limitations of direct 3D approaches etc.

**Strengths And Weaknesses:**

Strengths:
(+) The paper follows an interesting line of work fusing multiple representations of 3D pointclouds for scene understanding. Most interestingly, these representations are projected from the input pcl as pre-processing and do not stem directly from some input sensor, which raises an important gap in our understanding of pcl processing where operating directly on the input isn't as capable.
(+) Performance improvement over baselines is impressive.
(+) Geometric Flow Module is well explained and seems easily reproducible.

Weaknesses:
(-) The two conv branches are named "symmetric branches" and I wonder what symmetry is used here? Are the conv weights shared? What is done with respect to the difference in input images size? it seems both branches use ResNet34 so i'd assume the difference in image sizes persists throughout the network? maybe indicate that in the figure?
(-) Further, I'm missing an explanation of why should mu and theta be shared between the two branches? These are convolution layers that operate on very different input images.
(-) It seems BEV is only better than RV in the truck category and anywhere else performs worse, and often much worse. I therefore wonder whether BEV really contributes or would it have sufficed to augment the 3D network (kpconv) with just the RV branch? To my understanding, the reported ablation does not evaluate this since when the BEV branch is deactivated (lambda =  [0, 2, 0, 1, 0]) also alpha == 0 so GFNet isn't active. More generally, this may indicate that these two representations are far from optimal, and potentially other projections of the input pcl are better. It would be interesting to have at least a discussion about that. Also, given that these projections are produced artificially by the authors as preprocessing and not naturally emerge from a sensor, it's worth discussing why current network architectures cannot reproduce the same performance.
(-) The paper only tackles segmentation. Given how e.g. MV3D tested different fusion approaches, it will be interesting to adopt the geometric flow and compare it on detection tasks.
(-) The paper only tackles outdoor scenes: indoor scenes have many details along the height axes, therefore a common criticism against BEV is that it may lose these important details necessary for indoor tasks. Here, complementary features coming from RV and finally from 3D KPConv therefore I think it's important to know whether these projections are also useful for indoor scenes. In particular, KPConv was evaluated on Scannet so seeing whether projected views can assist its performance would be insightful.
(-) input features seem quite simplistic. For example, MV3D computed density and height maps for the BEV input features. Can the authors try this here as well?
(-) Adding image modality seems natural and would further strengthen the work


Minor:
(-) If Phi ranges from f_down to f_up should v tilde be: (phi - f_down) / (f_up - f_down) * H ?
(-) In Eq (3) polar is defined for (x,y) but in Eq (4) it is used with x,y,z input. How should the 3D input be understood?
(-) Would be useful to mention why after transforming the cartesian coordinates to polar, both are concatenated.
(-) in Attention Fusion computation, is Softmax computed globally on the entire image?
(-) Note that BEV and RV are indicated by subscripts throughout the paper notation (e.g. Qr and Qb) but suddenly in the prediction the notation becomes Rc and Bc. I suggest keeping consistency with the rest of the work.
(-) Further, under the current notation I believe the second loss term in Eq (9) should compare Bc with Qb and not Rb.
(-) grid sampling is not explained: is this mapping from HxW to original points N?
(-) bottom right illustration in figure 3 is confusing -- it's not clear where it fits in the large module. Seems to just repeat the output operations already listed? Also, the comma separated Rc and Bc notation is not explained. It should be clear they are not concatenated, and each has its own different H and W sizes and not share the same size.
(-) L_CL is a combination of losses. I assume weighted? What are the weights?
(-) how are the projected annotations Q_r and Q_b computed? what happens in projected view pixels that have points mapped to them with different classes?
(-) what is the ASPP module mentioned?
(-) what is meant by linear scaling of the features  "according to the resolution"?


Typos:
(-) In section 3.2: "we aims"
(-) not a typo but "generate probabilistic maps" sounds vague. Do the authors mean semantic class probability here? maybe spell it out more explicitly.
(-) kpconv in conclusion (fix capitalization)

---

> ### Author Response · Authors · 2022-08-19
> **Responses**
>
> Thanks for your valuable comments/suggestions! We will respond to your concerns in a point-to-point manner as follows.
>
> **Q1**: *The two conv branches are named "symmetric branches" and I wonder what symmetry is used here? Are the conv weights shared? What is done with respect to the difference in input images size? it seems both branches use ResNet34 so i'd assume the difference in image sizes persists throughout the network? maybe indicate that in the figure?*
>
> **A1**: The “symmetric” indicates that both RV and BEV have the same backbone and geometric flow architecture, but they do not share weights. We have removed this “symmetric” description to avoid confusion in the revision.
>
> Yes, both branches use ResNet34 and persist different images size throughout the entire network. The transformation matrix computation between RV and BEV has already considered different RV and BEV sizes. Specifically, the value in the transformation matrix is a ratio (normalized to [0, 1]), and it fetches the corresponding position by multiplying with the current specific size.
>
> **Q2**: *Further, I'm missing an explanation of why should mu and theta be shared between the two branches? These are convolution layers that operate on very different input images*
>
> **A2**: We expect to use mu and theta to represent that both RV and BEV use the same architecture, but they actually do not share weights. We have modified Figure 4 and its caption to show that they only have the same architecture but not the same weights in the revision.
>
> **Q3**: *It seems BEV is only better than RV in the truck category and anywhere else performs worse, and often much worse. I therefore wonder whether BEV really contributes or would it have sufficed to augment the 3D network (kpconv) with just the RV branch? To my understanding, the reported ablation does not evaluate this since when the BEV branch is deactivated (lambda = [0, 2, 0, 1, 0]) also alpha == 0 so GFNet isn't active. More generally, this may indicate that these two representations are far from optimal, and potentially other projections of the input pcl are better. It would be interesting to have at least a discussion about that. Also, given that these projections are produced artificially by the authors as preprocessing and not naturally emerge from a sensor, it's worth discussing why current network architectures cannot reproduce the same performance*
>
> **A3**: As shown in Figure 3, each branch (NxC) goes through a kpconv layer, and then concatenates features (Nx2C) from both branches, followed by another kpconv layer to learn how to fuse the features from RV and BEV for the same point (NxC). Thus, though lambda = [0, 2, 0, 1, 0] deactivates BEV and the kpconv for fusion, it actually does activate the kpconv layer in the RV branch, which evaluates the 3D network (kpconv) with just the RV branch and gets iou=61.0. After incorporating the BEV branch, it is improved to 63.0, which shows the effectiveness of BEV.
>
> We think the possible reason is that the well-developed 2D convolution networks (like ResNet) help to effectively extract corresponding discriminative features. For the methods that directly take the raw points from the lidar sensor as input, they have theoretically upper boundary accuracy (or iou) compared to other types of methods. However, there is still no powerful and unified architecture (such as ResNet in 2D images) for processing raw points and it is still under development. For example, the sampling strategy is different between RandLa-Net [8] (random sampling) and KPConv [7] (furthest point sampling).
>
> **Q4**: *The paper only tackles segmentation. Given how e.g. MV3D tested different fusion approaches, it will be interesting to adopt the geometric flow and compare it on detection tasks.*
>
> **A4**: The proposed geometric flow module design might not be suitable for object detection due to the following reasons: 1) The geometric flow module is to transform and fuse features in a pixel-to-pixel way between RV and BEV, originally and specially designed for semantic segmentation which is a fine-grained and per-pixel aware task or can be viewed as a per-pixel classification task. But for object detection, it requires using rectangles to locate those interested regions, thus it is more aware of global or patch-region information instead of pixel-degree features; 2) In GFNet, we only incorporate geometric flow module into each layer of the decoder (i.e., up-sampling layers). However, the decoder architecture (up-sampling layers) is usually not included in the object detection framework, making directly employing the geometric flow module not applicable. Therefore, we prefer to explore our designs in 3D object detection as future work.

---

> > ### Author Response · Authors · 2022-08-19
> > **Continuing Responses**
> >
> > **Q5**: *The paper only tackles outdoor scenes: indoor scenes have many details along the height axes, therefore a common criticism against BEV is that it may lose these important details necessary for indoor tasks. Here, complementary features coming from RV and finally from 3D KPConv therefore I think it's important to know whether these projections are also useful for indoor scenes. In particular, KPConv was evaluated on Scannet so seeing whether projected views can assist its performance would be insightful.*
> >
> > **A5**: Both both RV and BEV are not applicable to indoor scenes. For BEV, it looks points from top to down. However, indoor scenes usually include the ceiling, so when doing BEV projection, the ceiling points will cover all other below and meaningful points, leading to meaningless projections. Also, the objects in indoor scenes like the bookcase, table and chair, are distinguishable and meaningful in the vertical direction. But in BEV, the height information is lost, so these objects will be hardly recognized.
> >
> > As for RV, please refer to **RV under Indoor and Outdoor Scenes, Appendix** in the revision for a more detailed and clear explanation with figures. In short, point clouds in outdoor and indoor scenes are obtained in different ways. For example, SemanticKITTI is collected via a Velodyne HDL-64E lidar on the top of the driving car, which launches lasers to all-around (360) horizontal directions and to a certain degree [-25, 3] vertical directions. When applied RV projection, a meaningful cylindrical image can be obtained (please refer to Figure 1). But for the indoor dataset like S3DIS [1], it scans the entire room with a Matterport [2] scanner to generate point clouds. Besides, those indoor objects are also high-density. If we still want to use RV projection, it should lead to a severe distort image. We have also made some attempts using RV projection for S3DIS, but obtained meaningless images. That is why previous projection-based work [3,4,5] only employs RV projection on outdoor lidar-collected point clouds. As for indoor datasets like S3DIS, the main methods [6,7] usually take raw points as input directly because their size is much smaller than outdoor scenes.
> >
> > **Q6**: *input features seem quite simplistic. For example, MV3D computed density and height maps for the BEV input features. Can the authors try this here as well?*
> >
> > **A6**: As you can see from Eq.4, we have already taken z values of 3D points as an input feature, and sent to the network. In addition, as we stated in Sec.4.2 (Implementation Details), we use a polar grid size of *[480, 360, 32]* to cover a space of *[radius: (3m, 50m), z: (-3m,1.5m)]* relative to the lidar sensor. The grid first goes through a mini PointNet [6] to obtain the maximum feature activations along the *z* axis, leading to a reduced resolution *[480, 360]* for the BEV branch. Therefore, the height maps are essentially included as the input features.
> >
> > As for density, we follow the same rule as MV3D to use min(1.0, log(N+1)/log64) as the density feature, along with the original 9 features as the new input features for training. It achieves 62.5% mIoU, closed to our 63.0% performance. Note that this result is obtained using 8 v100-32G GPUS with batch 24 instead of 4 A100-40G GPUS with batch size 16 because we can not access to A100 anymore.  We suspect that the segmentation task is less sensitive to density than detection. For example, in detection, the bounding boxes for objects are very likely in density-changing locations. Besides, using min(1.0, log(N+1)/log64) to construct the density map might be suboptimal. The results for all classes are listed as follows:
> > |          class | mIoU      |
> > |            :-- |       --: |
> > |            car | 93.169707 |
> > |        bicycle | 50.922447 |
> > |     motorcycle |  61.86499 |
> > |          truck | 64.701611 |
> > |  other-vehicle | 33.455506 |
> > |         person | 72.010177 |
> > |      bicyclist | 87.437749 |
> > |   motorcyclist |       0.0 |
> > |           road | 95.700997 |
> > |        parking | 48.647264 |
> > |       sidewalk | 83.615088 |
> > |   other-ground |  0.025775 |
> > |       building | 90.997487 |
> > |          fence | 59.045076 |
> > |     vegetation | 88.108617 |
> > |          trunk | 67.824364 |
> > |        terrain | 75.055206 |
> > |           pole | 65.322179 |
> > |   traffic-sign | 48.877925 |
> > |           **avg** | **62.462217**|

---

> > > ### Author Response · Authors · 2022-08-19
> > > **Continuing Responses**
> > >
> > > **Q7**: *Adding image modality seems natural and would further strengthen the work*
> > >
> > > **A7**: The proposed geometric flow module is for pixel-to-pixel feature transformation and fusion. For nuScenes, it has all-around cameras to catch corresponding images for lidar points. Thus it is applicable to employ the geometric flow module to fuse image pixels and projected view pixels. But for SemanticKITTI, it only has the front camera, so it can not build a complete pixel-to-pixel mapping between images and projection views. A possible solution is using knowledge distillation to distill rich semantic information from images and fuse it to projection views. We prefer to leave it as future work.
> > >
> > > **Q8**: *If Phi ranges from f_down to f_up should v tilde be: (phi - f_down) / (f_up - f_down) * H ?*
> > >
> > > **A8**: If v = [(phi - f_down) / (f_up - f_down) ]*H, then when phi reaches f_up, v=H, which is not consistent to human sense. Because f_up represents the highest points collected by lidar, which should locate at the top of projection images. So a small modification need to be done: v = [1 - (phi - f_down) / (f_up - f_down) ]*H. As for the Eq.2 in our paper, it should be v = [(f_up - phi) / (f_up - f_down) ]*H, which is actually the same with  v = [1 - (phi - f_down) / (f_up - f_down) ]*H. We have fixed this issue in the revision.
> > >
> > > **Q9**: *In Eq (3) polar is defined for (x,y) but in Eq (4) it is used with x,y,z input. How should the 3D input be understood?*
> > >
> > > **A9**: The polar transformation is only for (x, y), and z is not changed. So *polar(x, y, z) = [polar(x, y), z]*, which is actually the cylindrical coordinate representation. So we have modified Eq.4 to *cylindrical(x,y,z) = [polar(x,y), z]* for clearer description in the revision.
> > >
> > > **Q10**: *Would be useful to mention why after transforming the cartesian coordinates to polar, both are concatenated.*
> > >
> > > **A10**: We follow PolarNet setting for a fair comparison to use 9 features to describe each point, including 3 relative distances, 3 polar coordinates, 2 cartesian coordinates (x,y) and 1 remission. PolarNet also does not have any ablation study or explanations for this setting. We suspect one potential reason is that the network can theoretically learn those cartesian information through polar coordinates, but explicitly feeding cartesian coordinates can help the network learning.
> > >
> > > We also make an attempt that removes 2 cartesian coordinates (x,y) from the input features for training. The experimental results are listed as follows. The results indicate the above-mentioned help is limited.
> > >
> > >  |        classes |    mIoU |
> > >  | --------------|----------  |
> > >  |            car | 94.884408 |
> > >  |        bicycle |  46.54603 |
> > >  |     motorcycle |  62.26939 |
> > >  |          truck |  72.18014 |
> > >  |  other-vehicle | 39.650762 |
> > >  |         person | 70.285022 |
> > >  |      bicyclist | 87.125736 |
> > >  |   motorcyclist |       0.0 |
> > >  |           road | 95.766741 |
> > >  |        parking | 46.332213 |
> > >  |       sidewalk | 83.380103 |
> > >  |   other-ground |  0.319497 |
> > >  |       building | 89.219022 |
> > >  |          fence | 54.456174 |
> > >  |     vegetation | 88.268191 |
> > >  |          trunk | 68.739742 |
> > >  |        terrain | 75.711268 |
> > >  |           pole | 65.707713 |
> > >  |   traffic-sign | 44.915348 |
> > >  |  **avg** | **62.408286** |
> > >
> > > **Q11**: *in Attention Fusion computation, is Softmax computed globally on the entire image?*
> > >
> > > **A11**: We only do Softmax on channel dimension. If the input feature is NxCxHxW, then softmax = torch.nn.Softmax(dim=1) is applied.
> > >
> > > **Q12**: *Note that BEV and RV are indicated by subscripts throughout the paper notation (e.g. Qr and Qb) but suddenly in the prediction the notation becomes Rc and Bc. I suggest keeping consistency with the rest of the work.*
> > >
> > > **A12**: Thanks for your suggestion, and we have modified these notions from R_c, B_c, R_3D, B_3D, F_3D to M_r, M_b, F_r, F_b, F_f for consistency in the revision.
> > >
> > > **Q13**: *Further, under the current notation I believe the second loss term in Eq (9) should compare Bc with Qb and not Rb.*
> > >
> > > **A13**: Yes. We have fixed this issue in the revision.
> > >
> > > **Q14**: *grid sampling is not explained: is this mapping from HxW to original points N?*
> > >
> > > **A14**: Yes, grid sampling is an operation that samples corresponding features from conv maps (HxWxC) to original points features (NxC) according to previous projection relationships. We actually expect to use the bottom right illumination in Figure 3 to show this in detail, but it seems to need a more illustrative explanation. We have modified this part of Figure 3 in the revision.

---

> > > > ### Author Response · Authors · 2022-08-19
> > > > **Continuing Responses**
> > > >
> > > > **Q15**: *bottom right illustration in figure 3 is confusing -- it's not clear where it fits in the large module. Seems to just repeat the output operations already listed? Also, the comma separated Rc and Bc notation is not explained. It should be clear they are not concatenated, and each has its own different H and W sizes and not share the same size.*
> > > >
> > > > **A15**: The original purpose of this subplot is to illustrate how the grid sampling and kpconv layer work. It just repeats certain operations in the main framework with extra annotated dimension information. The comma indicates both R_c and B_c have the same process instead of concatenation. We have modified this subplot in Figure 3 to be more illustrative in the revision.
> > > >
> > > > **Q16**: *L_CL is a combination of losses. I assume weighted? What are the weights?*
> > > >
> > > > **A16**:  L_CL is a combination of the cross entropy loss and the Lovasz-Softmax loss with weights 1:1. We have added this information in the revision.
> > > >
> > > > **Q17**: *how are the projected annotations Q_r and Q_b computed? what happens in projected view pixels that have points mapped to them with different classes?*
> > > >
> > > > **A17**: The projection matrix defines the correspondence between k_th point and pixel (u, v). When we have this matrix, we can assign the xyz coordinates and label of k_th point to pixel (u,v), which can lead to (H_r, W_r) for Q_r and (H_b, W_b) for Q_b.
> > > >
> > > > As we stated in **Geometric Alignment** in Sec.3.3 (between Eq.5 and Eq.6), if multiple points (different classes or not) project to the same pixel, the point with a smaller range is kept.
> > > >
> > > > **Q18**: *what is the ASPP module mentioned?*
> > > >
> > > > **A18**:  ASPP is Atrous Spatial Pyramid Pooling in [9]. The ASPP is usually used to explore multi-scale semantic information in the segmentation task. In our implementation, we use three different dilation rates in convolution with extra avg pooling to enhance multi-scale semantic information.
> > > >
> > > > **Q19**: *what is meant by linear scaling of the features "according to the resolution"?*
> > > >
> > > > **A19**: The value in the transformation matrix is actually a ratio (normalized to *[0, 1]*), and it fetches the corresponding position by multiplying with the specific resolution to get the feature.
> > > >
> > > > **Q20**: *Typos: (-) In section 3.2: "we aims" (-) not a typo but "generate probabilistic maps" sounds vague. Do the authors mean semantic class probability here? maybe spell it out more explicitly. (-) kpconv in conclusion (fix capitalization)*
> > > >
> > > > **A20**: Yes, it indicates ''the probability maps for each semantic class”. For this statement and other typos, we have modified them in the revision.
> > > >
> > > > **References**:
> > > >   1. Armeni et al. 3d semantic parsing of large-scale indoor spaces. CVPR 2016.
> > > >   2. Matterport. Matterport 3d models of interior spaces. http://matterport.com/, 2015. Accessed: 2015-06-01.
> > > >   3. Milioto et al. Rangenet++: Fast and accurate lidar semantic segmentation. IROS 2019.
> > > >   4. Wu et al. Squeezeseg: Convolutional neural nets with recurrent crf for real-time road-object segmentation from 3d lidar point cloud. ICRA 2018.
> > > >   5. Cortinhal et al. Salsanext: Fast, uncertainty-aware semantic segmentation of lidar point clouds. ISVC 2020.
> > > >   6. Qi et al. Pointnet: Deep learning on point sets for 3d classification and segmentation. CVPR 2017.
> > > >   7. Thomas et al. Kpconv: Flexible and deformable convolution for point clouds. ICCV 2019.
> > > >   8. Hu et al. Randla-net: Efficient semantic segmentation of large-scale point clouds. CVPR 2020.
> > > >   9. Chen et al. Rethinking atrous convolution for semantic image segmentation. ArXiv 2017

---

### Review · Reviewer_A5Ne · 2022-07-28

**Summary Of Contributions:**

This paper explores the geometric correspondence between different views in 3D point clouds. Experiments on  KITTI and nuScenes datasets show promising results.

**Broader Impact Concerns:**

No Broader Impact Concerns.

**Requested Changes:**

refer to Strengths And Weaknesses.

**Strengths And Weaknesses:**

1. 3D point cloud segmentation has indeed always been a fundamental task for point cloud related tasks. In this paper, it is feasible to use the projection of range view and bird's-eye view to complement each other, but the point clouds obtained from these two perspectives seem to have overlapping gaps, so how does this paper consider this problem? On the other hand, the translation from one perspective to another in this paper goes through three steps, and there seems to be a large loss of information in between.

2. The point cloud segmentation task actually serves the subsequent downstream tasks in more cases. The method in this paper seems to have achieved a certain gain, so how much does this gain help the downstream tasks? For example, the task of 3D object detection.

3. The methods proposed in this paper are all outdoor point cloud segmentation, so is it applicable to point cloud segmentation for indoor objects? If applicable, how did it perform? If not applicable, please explain why?

4. The experimental results in this paper are performed on the KITTI and nuScenes datasets, so what is the performance on the larger dataset Waymo? This dataset seems to be more suitable for evaluating the tasks related to this paper.

---

> ### Author Response · Authors · 2022-08-19
> **Responses**
>
> Thanks for your valuable comments/suggestions! We will respond to your concerns in a point-to-point manner as follows.
>
> **Q1**: *3D point cloud segmentation has indeed always been a fundamental task for point cloud related tasks. In this paper, it is feasible to use the projection of range view and bird's-eye view to complement each other, but the point clouds obtained from these two perspectives seem to have overlapping gaps, so how does this paper consider this problem? On the other hand, the translation from one perspective to another in this paper goes through three steps, and there seems to be a large loss of information in between.*
>
> **A1**: Yes, there are overlapping points. Actually, for each point, it will have two feature representations separately from RV and BEV. We concatenate these features and feed it to the next kpconv layer. Specifically, as shown in Figure 3, we first apply the grid sample on both RV and BEV branches according to the projection relationships to obtain feature for each point (i.e., R3D and B3D). We then concatenate R3D and B3D and feed it to the kpconv layer, making kpconv to learn how to fuse these features.
>
> As for the transformation between two views, we build a one-to-one transformation matrix in eq.5-8, sec.3.3. So for each element in one view, we can always fetch a theoretically-correct corresponding element from another view via this matrix, and then fuse their features. A potential loss comes from one 2D position projected by multiple 3D points, so when fetching feature from this type of position, it might be not what we expect. To handle these cases, we design the Attention Fusion (Please refer to Algorithm 1) to selectively fuse those features.
>
> **Q2**: *The point cloud segmentation task actually serves the subsequent downstream tasks in more cases. The method in this paper seems to have achieved a certain gain, so how much does this gain help the downstream tasks? For example, the task of 3D object detection.*
>
> **A2**: The proposed geometric flow module design might not be suitable for object detection due to the following reasons: 1) The geometric flow module is to transform and fuse features in a pixel-to-pixel way between RV and BEV, originally and specially designed for semantic segmentation which is a fine-grained and per-pixel aware task or can be viewed as a per-pixel classification task. But for object detection, it requires using rectangles to locate those interested regions, thus it is more aware of global or patch-region information instead of pixel-degree features; 2) In GFNet, we only incorporate geometric flow module into each layer of the decoder (i.e., up-sampling layers). However, the decoder architecture (up-sampling layers) is usually not included in the object detection framework, making directly employing the geometric flow module not applicable. Therefore, we prefer to explore our designs in 3D object detection as future work.

---

> > ### Author Response · Authors · 2022-08-19
> > **Continuing Responses**
> >
> > **Q3**: *The methods proposed in this paper are all outdoor point cloud segmentation, so is it applicable to point cloud segmentation for indoor objects? If applicable, how did it perform? If not applicable, please explain why?*
> >
> > **A3**: Both both RV and BEV are not applicable to indoor scenes. For BEV, it looks points from top to down. However, indoor scenes usually include the ceiling, so when doing BEV projection, the ceiling points will cover all other below and meaningful points, leading to meaningless projections. Also, the objects in indoor scenes like the bookcase, table and chair, are distinguishable and meaningful in the vertical direction. But in BEV, the height information is lost, so these objects will be hardly recognized.
> >
> > As for RV, please refer to **RV under Indoor and Outdoor Scenes, Appendix** in the revision for a more detailed and clear explanation with figures. In short, point clouds in outdoor and indoor scenes are obtained in different ways. For example, SemanticKITTI is collected via a Velodyne HDL-64E lidar on the top of the driving car, which launches lasers to all-around (360) horizontal directions and to a certain degree [-25, 3] vertical directions. When applied RV projection, a meaningful cylindrical image can be obtained (please refer to Figure 1). But for the indoor dataset like S3DIS [1], it scans the entire room with a Matterport [2] scanner to generate point clouds. Besides, those indoor objects are also high-density. If we still want to use RV projection, it should lead to a severe distort image. We have also made some attempts using RV projection for S3DIS, but obtained meaningless images. That is why previous projection-based work [3,4,5] only employs RV projection on outdoor lidar-collected point clouds. As for indoor datasets like S3DIS, the main methods [6,7] usually take raw points as input directly because their size is much smaller than outdoor scenes.
> >
> > Overall, both RV and BEV are not suitable for indoor scenes.
> >
> > **Q4**: *The experimental results in this paper are performed on the KITTI and nuScenes datasets, so what is the performance on the larger dataset Waymo? This dataset seems to be more suitable for evaluating the tasks related to this paper.*
> >
> > **A4**: The datasets we evaluate in the paper, SemanticKITTI and nuScenes, are the two most widely used large-scale benchmarks in lidar point cloud segmentation field. Following many previous classical work [6,7,8], our reported experimental results on these two datasets are representative and convincing.
> >
> > We notice that Waymo Open Dataset released the corresponding 3D semantic segmentation labels in a very recent time (March 2022), and subsequently improved the quality of these labels in May 2022. Meanwhile, it is a too large dataset given our current computational resources, and we prefer to leave it for future work.
> >
> >
> > **References**:
> >   1. Armeni et al. 3d semantic parsing of large-scale indoor spaces. CVPR 2016.
> >   2. Matterport. Matterport 3d models of interior spaces. http://matterport.com/, 2015. Accessed: 2015-06-01.
> >   3. Milioto et al. Rangenet++: Fast and accurate lidar semantic segmentation. IROS 2019.
> >   4. Wu et al. Squeezeseg: Convolutional neural nets with recurrent crf for real-time road-object segmentation from 3d lidar point cloud. ICRA 2018.
> >   5. Cortinhal et al. Salsanext: Fast, uncertainty-aware semantic segmentation of lidar point clouds. ISVC 2020.
> >   6. Qi et al. Pointnet: Deep learning on point sets for 3d classification and segmentation. CVPR 2017.
> >   7. Thomas et al. Kpconv: Flexible and deformable convolution for point clouds. ICCV 2019.
> >   8. Hu et al. Randla-net: Efficient semantic segmentation of large-scale point clouds. CVPR 2020.

---

### Review · Reviewer_bVo2 · 2022-08-12

**Summary Of Contributions:**

GFNet (geometric flow network) addresses the critical problem of 3D Semantic segmentation on point clouds.  GFNet explores the geometric correspondence between different views in an align-before-fuse manner, in contrast with traditional methods that use late fusion.  This new method achieves state-of-the-art compared to projection methods.

**Broader Impact Concerns:**

The article has not had a broader impact statement. This method can be used for autonomous driving or can also be used for military purposes.

**Requested Changes:**

I miss a section explaining the computational needs of the method for training and inference. It says that it works at 10FPS but I didn't see the computation requirements for that. Moreover, compare it with the speed of AMVNet which has almost the same mIOU.

I miss an analysis of the weak points of GFNet and an agenda of future work on how to improve it.

I miss some more recent references.

**Strengths And Weaknesses:**

The article is well written, easy to follow, and has adequate references. The literature review is comprehensive, well organized, and up-to-date. I may suggest, however, including some more recent references.

Figures 1 and 2 help understand the method and Figure 3 clearly shows the architecture. GFNet is novel, simple, and end-to-end differentiable. The notation seems correct and adequate.

The experiments are conducted in relevant datasets and evaluated using standard metrics. The results outperform the state-of-the-art in all the cases. However, the results are not much better than these of AMVNet. Nonetheless, GFNet is easier to train than AMVNet. The ablation studies are complete.

---

> ### Author Response · Authors · 2022-08-19
> **Responses**
>
> Thanks for your valuable comments/suggestions! We will respond to your concerns in a point-to-point manner as follows.
>
> **Q1**: *I miss a section explaining the computational needs of the method for training and inference. It says that it works at 10FPS but I didn't see the computation requirements for that. Moreover, compare it with the speed of AMVNet which has almost the same mIOU.*
>
> **A1**: Our models are trained and tested with AMD EPYC 7742 64-Core Processor and Nvidia A100-SXM4-40GB GPU. The GPU info has already been described in Sec.4.2 (Implementation Details), and we also added the CPU information in the revision. In addition, both our source code and pretrained models will be made publicly available.
>
> As for AMVNet, it does not report the FPS in the paper, while the source code is also not available. In our project page, we will be happy to make comparisons with more methods in terms of FPS if the source code and models are publicly available.
>
> **Q2**: *I miss an analysis of the weak points of GFNet and an agenda of future work on how to improve it.*
>
> **A2**: GFNet achieves similar mIoU compared to AMVNet, but clearly outperforms AMVNet under Accuracy (92.4% vs 91.3%) in SemanticKITTI and FW IoU (90.4% vs 89.5%) in nuScenes. It reveals that GFNet performs much better on frequent classes while somewhat struggles on those rare/small classes. Possible solutions to alleviate this limitation might be frequency-weighted data sampling, stronger data augmentation, and frequency-weighted loss designs. We have added the above discussion to Sec.4.5 (Comparison with Recent State-of-the-Arts) in the revision.
>
> **Q3**: *I miss some more recent references.*
>
> **A3**: For projection-based methods, all recent state-of-the-art methods have been reviewed, and a detailed comparison between them and GFNet is also provided. For other types of methods, we have discussed several representative methods. Since many recent methods mainly focus on hybrid methods, i.e., simultaneously using multiple formats (voxel, points, and natural images), we review them in an additional subsection, **Hybrid Methods** of Sec.2 (Related Work), in the revision.

---

### Decision · Action_Editors · 2022-09-16

**Recommendation:** Accept with minor revision

**Comment:**

Two reviewers recommend acceptance (m9mP: Accept; bVo2: Leaning Accept) and one rejection (A5Ne: Leaning Reject). While the authors' responses addressed most of the initial reviewers' concerns, some questions raised by A5Ne remained open, such as the impact on downstream tasks (e.g., 3D object detection), the results on the larger Waymo dataset, and the applicability to indoor scenes. The Action Editor nonetheless believes that there is sufficient material for acceptance, as acknowledged by the other two reviewers. However, the authors are encouraged to incorporate their answers regarding 3D object detection and indoor scenes as limitations of the approach.

---

> ### Author Response · Authors · 2022-09-21
> **Camera-Ready Version**
>
> Dear reviewers and AE
>
> Thanks for your precious time and comments/suggestions in the review process, which helped us to further improve the quality of the paper. We have uploaded the deanonymized camera-ready version.